# ProEval: Proactive Failure Discovery and Efficient Performance Estimation for Generative AI Evaluation

**Yizheng Huang** [* 1]  **Wenjun Zeng** [1]  **Aditi Kumaresan** [1]  **Zi Wang** [* 1]

## Abstract

Evaluating generative AI models is increasingly resource-intensive due to slow inference, expensive raters, and a rapidly growing landscape of models and benchmarks. We propose PROEVAL, a proactive evaluation framework that leverages transfer learning to efficiently estimate performance and identify failure cases. PROEVAL employs pre-trained Gaussian Processes (GPs) as surrogates for the performance score function, mapping model inputs to metrics such as the severity of errors or safety violations. By framing performance estimation as Bayesian quadrature (BQ) and failure discovery as superlevel set sampling, we develop uncertainty-aware decision strategies that actively select or synthesize highly informative inputs for testing. Theoretically, we prove that our pre-trained GP-based BQ estimator is unbiased and bounded. Empirically, extensive experiments on reasoning, safety alignment, and classification benchmarks demonstrate that PROEVAL is significantly more efficient than competitive baselines. It requires 8–65x fewer samples to achieve estimates within $\pm 1\%$ of the ground truth, while simultaneously revealing more diverse failure cases under a stricter evaluation budget. Our open-sourced code and data can be found at https://github.com/google-deepmind/proeval.

## 1. Introduction

Evaluation of modern generative AI models has become a critical backbone for model selection (Chen et al., 2023), assurance of safety and fairness (Aroyo et al., 2023; Zhang et al., 2024b), discovery of capabilities (Srivastava et al.,

---

[*]Equal contribution  [1]Google DeepMind. Correspondence to: Yizheng Huang <yizhengh@google.com>, Zi Wang <wangzi@google.com>.

*Proceedings of the 43$^{rd}$ International Conference on Machine Learning*, Seoul, South Korea. PMLR 306, 2026. Copyright 2026 by the author(s).

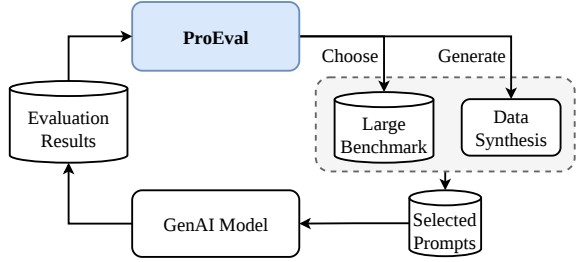

*Figure 1.* Overview of the PROEVAL framework. It acts as an active selector that continuously updates itself based on past evaluation results. To test the GenAI model efficiently, it dynamically builds a set of selected prompts by either choosing test examples from a large benchmark or performing data synthesis to discover specific model failures.

2023), measurement of intelligence (Phan et al., 2025), and more. As these models are increasingly deployed in high-stakes environments, the demand for rigorous assessment has driven a proliferation of new benchmarks (NeurIPS, 2025) and evaluation strategies (Zheng et al., 2023; Zhang et al., 2024a; Guan et al., 2025).

Yet, the current paradigm of evaluation is becoming unsustainable and resource-intensive. Besides the cost of generation, reliable assessment often requires costly human raters or expensive LLM-based "judges". This computational burden is exacerbated by the sheer scale of modern benchmarks. Running comprehensive evaluations across multiple models and datasets can cost thousands of dollars and require days of compute time. Moreover, during GenAI model development and quality iteration, development teams must frequently run numerous evaluation tasks against the same benchmarks, even for the most minor changes at different scales. Consequently, researchers and practitioners often resort to downsampling test data (Kossen et al., 2021; Kipnis et al., 2025), which may yield less accurate estimates and fail to uncover rare but critical failure cases.

To address these inefficiencies, we introduce PROEVAL, a sample-efficient framework combining transfer learning and Bayesian modeling to accurately estimate performance and proactively identify failure cases (Figure 1). At its core, PROEVAL treats the performance score of a target model as an unknown function $f$ that maps an input $x$ to a metric,

such as the severity of an error or unsafe response. Instead of learning this function from scratch, we employ transfer learning to construct a highly informed Gaussian process (GP) prior, and we design active selection and synthesis strategies to strategically target which inputs to evaluate.

To construct the GP prior, PROEVAL exploits the structural correlations in model performance. For standard benchmarks, we derive the GP's covariance directly from historical evaluation results on other models, capturing how performance on one input is predictive of another. For new domains or modalities, we construct an encoder based on off-the-shelf pre-trained embedding models (e.g., for text or image) to generate input features for the GP mean and kernel functions. We further pre-train the encoder parameters and the GP hyperparameters to fully capture the underlying performance correlations. This allows the GP surrogate to generalize based on semantic or visual similarity, and infer that a new input is likely to fail because it resembles known failure cases without performing an additional evaluation.

Building on this informed prior, PROEVAL constructs active data acquisition strategies and addresses two synergistic objectives within a single probabilistic framework: *performance estimation* to obtain an aggregate assessment, and *failure discovery* to zoom in on the specific input regions responsible for performance degradation.

First, PROEVAL formulates performance estimation as Bayesian quadrature (BQ). Instead of simple Monte Carlo averaging, BQ treats the integral $\int f(x)p(x)\,\mathrm{d}x$ as a random variable derived from the GP posterior. This allows us to analytically compute the variance of the performance estimate. By actively selecting data points that minimize this posterior variance, PROEVAL achieves high sample efficiency, reaching the true performance metric with far fewer samples than competitive baselines.

Simultaneously, PROEVAL tackles failure case discovery via superlevel set sampling, aiming to characterize the region $\{x \mid f(x) \geq \lambda\}$ where the model likely fails. To do this efficiently, we design an acquisition function that balances exploitation (targeting inputs where the predicted failure probability is high) with exploration (targeting inputs with high epistemic uncertainty). This ensures we identify failures that are both severe and diverse.

We further enhance failure discovery via active synthesis of new data. We employ an LLM-based generative approach that uses identified "hard" examples as in-context anchors to generate new, more challenging inputs. To prevent the generator from collapsing into a single failure mode, we introduce a bi-level sampling strategy. This approach first samples high-level semantic topics before generating specific test cases, forcing the system to uncover diverse failure patterns across the model's capabilities.

Theoretically, we prove that our BQ estimator based on pre-trained GPs is unbiased and bounded under mild assumptions. Empirically, PROEVAL demonstrates compelling efficiency in performance estimation, often reaching within 1% estimation error with only 1 to 27 evaluated inputs, significantly surpassing the performance of competitive baselines. For failure discovery, PROEVAL achieves about 2-5x higher failure detection rates and better diversity in the semantic space, compared to other LLM-based generation methods. Our extensive ablation studies show the critical roles played by our transfer learning approach, active sampling algorithms and LLM-enhanced failure discovery.

Our contributions are as follows: 1) A unified evaluation framework that grounds performance estimation in Bayesian quadrature and failure discovery in superlevel set sampling. 2) A flexible transfer learning mechanism for evaluation that constructs informed priors from either direct historical statistics or semantic embeddings. 3) Active sampling strategies tailored for sample-efficient performance estimation and an exploration-exploitation balanced strategy for identifying failure regions. 4) A hierarchical, topic-aware synthesis algorithm that leverages LLM-based in-context generation anchored on identified hard examples to actively synthesize and discover diverse failure modes. 5) The first theoretical proof that BQ with pre-trained GPs is unbiased and bounded without assuming a known GP prior. 6) Comprehensive empirical results demonstrating the remarkable sample efficiency and effectiveness of PROEVAL.

**Related work.** Detailed literature review can be found in §A. Existing efficient evaluation methods typically focus on benchmark pruning to identify subsets (Polo et al., 2024; Kipnis et al., 2025; Vivek et al., 2024) or active testing via surrogates (Kossen et al., 2021; 2022; Berrada et al., 2025), but these often lack model-specific adaptability or suffer from high-variance sampling. Similarly, current failure discovery and red teaming approaches are frequently limited by a reliance on manual specifications, fine-tuning, or sample inefficiency (Perez et al., 2022; Chao et al., 2024; Mehrotra et al., 2024; Samvelyan et al., 2024). PROEVAL departs from these disjointed methods by establishing a Bayesian framework with transfer learning that integrates performance estimation and failure discovery. By using a shared GP for both Bayesian quadrature and probabilistic superlevel set sampling, PROEVAL enables a unified, proactive, and sample-efficient approach to model evaluation.

## 2. Our Framework: Proactive Evaluation

We introduce PROEVAL, which frames *performance estimation* and *failure discovery* as dual Bayesian objectives (§2.1). The framework leverages transfer learning to construct strong GP priors (§2.2), enabling active sampling strategies for both estimation (§2.3) and discovery (§2.4).

## 2.1. Bayesian Formulation of Evaluation

We formalize model evaluation as a dual-objective problem over the input space $\mathcal{X}$. Let $f : \mathcal{X} \mapsto \mathbb{R}$ be the underlying performance score function (e.g., error severity), and $p(x)$ be the test distribution. We aim to use minimal evaluations of $f$ to estimate the global expected score $S$, and identify the superlevel set $\mathcal{X}_\lambda$ containing inputs where the model fails (score exceeds threshold $\lambda$). That is,

$$S = \int_{\mathcal{X}} f(x)p(x)\,\mathrm{d}x \quad \text{and} \quad \mathcal{X}_\lambda = \{x \mid f(x) \geq \lambda\}. \quad (1)$$

**Gaussian Process (GP) Surrogate.** To address these objectives sample-efficiently, we place a GP prior $f \sim GP(\mu, k)$. Given $t$ noisy observations $D_t = \{(x_\tau, y_\tau)\}_{\tau=1}^t$, where $y_\tau \sim \mathcal{N}(f(x_\tau), \sigma^2)$, the posterior $f \mid D_t \sim \mathcal{GP}(\mu_t, k_t)$, is given by

$$\mu_t(x) = \mu(x) + k(x, \boldsymbol{x}_t)K_t^{-1}(\boldsymbol{y}_t - \mu(\boldsymbol{x}_t)),$$
$$k_t(x, x') = k(x, x') - k(x, \boldsymbol{x}_t)K_t^{-1}k(\boldsymbol{x}_t, x'), \quad (2)$$

where vector $\boldsymbol{x}_t = [x_\tau]_{\tau=1}^t$, vector $\boldsymbol{y}_t = [y_\tau]_{\tau=1}^t$, $\mu(\boldsymbol{x}_t) = [\mu(x_\tau)]_{\tau=1}^t$, $k(x, \boldsymbol{x}_t) = [k(x, x_\tau)]_{\tau=1}^t$ and matrix $K_t = [[k(x_\tau, x_{\tau'})]_{\tau=1}^t]_{\tau'=1}^t + I\sigma^2$. The mean function $\mu$, kernel $k$ and noise variance $\sigma^2$ are unknown and will be learned via transfer learning.

**Linear kernel on embeddings.** As a special case, the GP can use a linear kernel defined over an encoder $\phi : \mathcal{X} \mapsto \mathbb{R}^d$, such that $k(x, x') = \phi(x)^{\mathrm{T}}\phi(x')$. Defining the embedding matrix as $Z = [\phi(x_\tau)]_{\tau=1}^t \in \mathbb{R}^{d \times t}$, the posterior evaluates to:

$$\mu_t(x) = \mu(x) + \sigma^{-2}\phi(x)^{\mathrm{T}}\widetilde{K_t}Z(\boldsymbol{y}_t - \mu(\boldsymbol{x}_t)),$$
$$k_t(x, x') = \phi(x)^{\mathrm{T}}\widetilde{K_t}\phi(x'), \quad (3)$$

where $\widetilde{K_t} = (ZZ^{\mathrm{T}}\sigma^{-2} + \boldsymbol{I})^{-1} \in \mathbb{R}^{d \times d}$. Notably, computing this inverse requires $O(d^3)$ operations, reducing the computational complexity from the $O(t^3)$ scaling seen in Equation (2). Furthermore, as the number of observations $t$ increases, $\widetilde{K_t}$ can be efficiently updated in $O(d^2)$ time using the Sherman-Morrison formula.

**Posteriors for evaluation objectives.** Once the GP posterior is established, we can use it to derive distributions over our evaluation objectives in Equation (1).

*Bayesian Quadrature (BQ):* By approximating the integral $S$ as a sum over finite test samples $\{x_j\}_{j=1}^M$ drawn *i.i.d.* from $p(x)$, we derive the following posterior mean and variance:

$$\mathbb{E}[S \mid D_t] \approx \frac{1}{M}\sum_{j=1}^M \mu_t(x_j), \ \mathbb{V}[S \mid D_t] \approx \frac{1}{M^2}\sum_{j,j'=1}^M k_t(x_j, x_{j'}). \quad (4)$$

*Probabilistic Superlevel Sets:* The probability that an input $x$ belongs to the failure region is $p(x \in \mathcal{X}_\lambda) = p(f(x) \geq \lambda \mid D_t)$. We approximate the region as

$$\mathcal{X}_\lambda^\beta = \{x \mid \mu_t(x) + \beta\sigma_t(x) \geq \lambda\}, \quad (5)$$

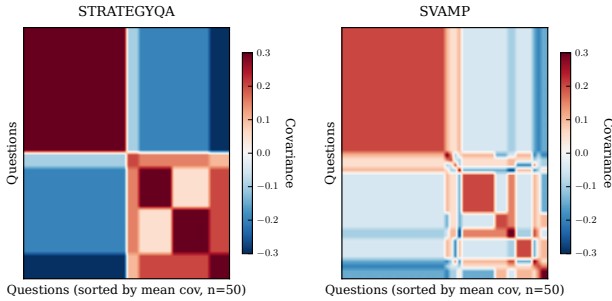

STRATEGYQA        SVAMP

*Figure 2.* **Empirical validation of performance correlations.** Sample covariance matrices of question performance on (left) StrategyQA and (right) SVAMP, computed across $N = 5$ language models (GPT-4o, Gemini 1.5 Pro, Gemma-7b/27b, and Qwen-72b). Questions are filtered to those with non-zero variance and ordered by mean covariance (top/bottom 25 shown). Red indicates positive covariance (models succeed/fail together); Blue indicates negative covariance. The distinct block structure reveals strong correlations between questions, supporting the validity of Assumption 1.

where $\sigma_t(x) = \sqrt{k_t(x, x)}$ is the posterior standard deviation and $\beta$ controls the confidence level.

## 2.2. Transfer Learning for GP Priors

The efficacy of PROEVAL hinges on the quality of the GP prior. We leverage the abundance of relevant historical evaluation data to construct this prior, formalized as follows:

**Assumption 1.** *Let $\mathcal{D} = \{D_i\}_{i=1}^N$ be a set of historical datasets, where $D_i = \{(x_{ij}, y_{ij})\}_{j=1}^{M_i}$. We assume the observations are perturbed with noise, $y_{ij} \sim \mathcal{N}(f_i(x_{ij}), \sigma^2)$, where the score functions $f_i$ are drawn from a shared GP prior $f_i \sim \mathcal{GP}(\mu, k)$ with unknown mean $\mu$, kernel $k$ and noise variance $\sigma^2$.*

This assumption posits that performance scores exhibit structural correlations across inputs. For example, questions testing similar reasoning skills often yield positive covariance (models tend to succeed or fail together), while distinct failure cases may result in negative covariance. Capturing these correlations is critical, as it allows PROEVAL to infer the performance on unobserved inputs based on the evaluation of correlated examples. Figure 2 empirically validates this assumption, revealing strong covariance structures in standard benchmarks.

Our goal here is to learn the GP from historical datasets to accurately model a new evaluation task with score function $f$. Note that the inputs to function $f$ are the same as inputs to GenAI models, often consisting of texts and images, and traditionally GPs do not use these modalities as inputs. A straightforward way to make GPs compatible with those inputs is to project the inputs into a finite-dimensional feature space. Depending on the available data, we introduce two distinct strategies to construct these features and learn the

GP.

### 2.2.1. SCORE FEATURES VIA EMPIRICAL STATISTICS

When evaluating a target model on a fixed target benchmark, we often have access to evaluation results from other models on the exact same set of questions. Under Assumption 2, this shared structure allows us to extract features directly from the historical score matrix.

**Assumption 2** (Standard Benchmarking). *All historical datasets evaluate the identical set of inputs $\{x_j\}_{j=1}^M$, such that $D_i = \{(x_j, y_{ij})\}_{j=1}^M$ for all $i \in \{1, \ldots, N\}$.*

Let $\boldsymbol{y}_i = [y_{ij}]_{j=1}^M$ be the vector of scores for model $i$ and matrix $Y = [\boldsymbol{y}_i]_{i=1}^N$. Under Assumption 1, the sample mean and covariance estimates are computed across the $N$ historical models:

$$\hat{\boldsymbol{u}} = \frac{1}{N} Y \times 1_N, \quad \hat{\Sigma} = \frac{1}{N-1}(Y - \hat{\boldsymbol{u}})(Y - \hat{\boldsymbol{u}})^{\mathrm{T}}. \quad (6)$$

Notably, these empirical statistics are equivalent to a GP prior with mean $\hat{\mu}(x_j) = \hat{\boldsymbol{u}}_j$ and linear kernel $\hat{k}(x, x') = \phi(x)^{\mathrm{T}}\phi(x')$. By defining the normalized score feature as $\phi(x_j) = \frac{1}{\sqrt{N-1}}[y_{ij} - \hat{\boldsymbol{u}}_j]_{i=1}^N$, the kernel exactly reconstructs $\hat{\Sigma}$.

Now we consider the GP posterior. We provide theoretical guarantees showing the posterior BQ estimator $\hat{S}_t = \frac{1}{M}\sum_{j=1}^M \hat{\mu}_t(x_j)$ based on this learned prior is unbiased and its deviation from the ground truth estimate $S_t = \frac{1}{M}\sum_{j=1}^M \mu_t(x_j)$ is bounded.

**Theorem 3** (Performance Estimation). *Assume $N \gg t$ and $\kappa \geq k(x, x)$. Given Assumption 1, 2 and $t$ observed scores on the target model, the performance estimator $\hat{S}_t$ satisfies*

$$\mathbb{E}[\hat{S}_t] = \mathbb{E}[S \mid D_t] \text{ and } |\hat{S}_t - S_t| \leq a'\sqrt{\kappa + \sigma^2}, \quad (7)$$

*where $a' = \left(\frac{4M\left(t+1+2\sqrt{t\log\frac{4M}{\delta}}+2\log\frac{4M}{\delta}-2/N\right)}{(N-t-2)\delta}\right)^{\frac{1}{2}}$, and the bound holds with probability $1 - \delta$.*

The proof can be found in §B. Our bound provides a worst-case guarantee and characterizes how error scales with the scale of the kernel ($\kappa$) and the number of models ($N$), offering the insight that increasing $N$ can lead to better estimation. The bound is non-vacuous if $N$ (the number of historical models) grows at a faster rate than $M$ (the number of test samples)[1]. Our empirical results in §3.2 demonstrate that even when the theoretical bound is loose, the estimation accuracy remains high.

---

[1]During model development with many evaluations on a validation set, it is likely that this $N > M$ condition is satisfied, because of the sweep over learning rate, batch size, architecture variants, regularizations, data scales, etc. Each configuration of those parameters corresponds to multiple models since there are many model checkpoints during training.

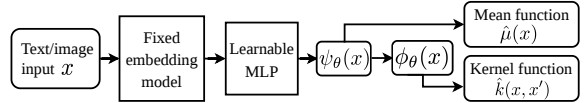

*Figure 3.* **Transfer learning via embeddings.** We use a pre-trained embedding model adapted via learnable parameters $\theta$ to define the GP and leverage input similarity even when historical data for the specific inputs is unavailable. To balance stability and flexibility, we use a mean function bounded by the scale of $\psi$ and an expressive Matérn kernel (see Section 2.2.2 for details).

### 2.2.2. PROMPT FEATURES VIA LEARNED EMBEDDINGS

To transfer knowledge across different benchmarks or when historical results are missing for specific inputs, Assumption 2 may not hold and we do not have direct access to score features. Instead, we construct a GP prior based on semantic similarity, using text (or any other input modality) embeddings to map inputs into a shared latent space.

There are many possible GP setups for how to use the embeddings. To imitate the use of score features, one option is to approximate the GP mean and kernel as $\hat{\mu}(x) = \frac{1}{d}\psi_\theta(x) \times 1_d$ and $\hat{k}(x, x') = \frac{1}{d-1}(\psi_\theta(x) - \hat{\mu}(x))^{\mathrm{T}}(\psi_\theta(x') - \hat{\mu}(x'))$, where $\psi_\theta : \mathcal{X} \mapsto \mathbb{R}^d$ is an embedding function (e.g., a transformer) with parameters $\theta \in \mathbb{R}^{d_\theta}$. Analogous to score features, this is equivalent to using a linear kernel with a centered encoder $\phi_\theta$:

$$\phi_\theta(x) = \frac{1}{\sqrt{d-1}}(\psi_\theta(x) - \frac{1}{d}\psi_\theta(x) \times 1_d) \in \mathbb{R}^d \quad (8)$$

Figure 3 illustrates this GP architecture. The linear formulation in Equation (8) allows us to inspect $\psi_\theta$ to ensure embedding magnitudes align with score ranges. However, the framework is flexible. Other choices include stationary kernels over embeddings; e.g., Matérn kernel $\hat{k}(x, x') = k_{\text{Matérn}}(\phi_\theta(x), \phi_\theta(x'))$, which we used in our experiments.

We optimize encoder parameters $\theta$ (including other GP hyperparameters) by maximizing the log-likelihood of the historical data across all $N$ datasets: $\hat{\theta} = \arg\max_\theta \sum_{i=1}^N \log p(\boldsymbol{y}_i \mid \theta)$. This optimization enables zero-shot generalization. Even for inputs never previously evaluated, the GP predicts difficulty based on semantic similarity to historical cases in the learned embedding space.

### 2.2.3. SELECTING SOURCE DATA

Regardless of whether the GP prior is constructed using empirical score features (Section 2.2.1) or learned prompt features (Section 2.2.2), its effectiveness relies on the relevance of the source data. Specifically, the validity of Assumption 1 depends on selecting historical datasets that align with the target evaluation. Using every available model to construct the prior assumes all models' score functions are samples from the same underlying distribution. This assumption fails when evaluating an out-of-distribution target model,

leading to negative transfer.

To automate and optimize this selection while preventing negative transfer, PROEVAL employs Gaussian Mixture Model (GMM) Clustering. We project different models' scores ($[y_{ij}]_{j=1}^M$) on a reference benchmark onto a lower-dimensional space using PCA and fit a GMM to identify models with similar behavioral profiles. By default, we use all available benchmarks except the target benchmark as this reference. The prior is then constructed exclusively from historical models within the target model's cluster. To maintain reliability, the framework uses an abstention rule: it abstains from estimation if the target's cluster contains fewer than three models, as sparse clusters indicate a lack of sufficient source data to form an informative prior.

Alternative selection heuristics, such as Spearman rank correlation or distance-based constraints (e.g., Mahalanobis distance), can be used when clustering is difficult. We provide a full empirical comparison and ablation of these design choices in Section E.

### 2.3. Active Performance Estimation

To efficiently estimate the performance integral $S$, we aim to minimize the estimator variance $\mathbb{V}[S \mid D_t]$ defined in Equation (4). We adopt a greedy acquisition strategy, selecting the next input $x_{t+1}$ that maximizes the reduction in posterior variance:

$$x_{t+1} = \arg\max_{x \in \mathcal{X}} \left( \mathbb{V}[S \mid D_t] - \mathbb{V}[S \mid D_t \cup x] \right) \quad (9)$$

Because this acquisition function is independent of the actual observations of the function $f$, it is possible to precompute batches of test inputs for parallel evaluation. For clarity, we outline the sequential formulation in Algorithm 1.

---

**Algorithm 1** PROEVAL for Performance Estimation

**Input:** Historical data $\mathcal{D}$ in Assumption 1, $D_0 = \emptyset$
Learn encoder $\phi$ according to Section 2.2 and set $\mu = \hat{\mu}$, $k = \hat{k}$
**for** $t = 0, \cdots, T - 1$ **do**
    Compute $x_{t+1}$ from Equation (9)
    $y_{t+1} \leftarrow \text{EVALUATE}(f, x_{t+1})$
    $D_{t+1} \leftarrow D_t \cup \{(x_{t+1}, y_{t+1})\}$
    Update performance estimate $\mathbb{E}[S \mid D_{t+1}]$ via Equation (4)
**end for**

---

### 2.4. Proactive Failure Case Discovery

Besides performance estimation, we aim to proactively identify failure cases by efficiently sampling from the failure superlevel set $\mathcal{X}_\lambda = \{x \mid f(x) \geq \lambda\}$ to discover diverse, high-severity inputs (e.g., safety violations or reasoning errors). We propose three strategies that build upon one another, progressing from efficient retrieval in static datasets to diversity-driven data synthesis.

**Strategy 1: Superlevel set sampling (SS).** This strategy aims to retrieve failure cases within a static pool of unlabeled

inputs $D_{\text{pool}}$. We design an acquisition function that, when maximized, targets inputs within the probable failure region, Equation (5), while ensuring high information gain:

$$\alpha_{\text{SS}}(x \mid D_t) = \mathbb{1}(\mu_t(x) + \beta\sigma_t(x) \geq \lambda) \times k_t(x, x). \quad (10)$$

The indicator term restricts the search to the probable failure set, where $\beta$ controls the confidence threshold (e.g., $\beta = 0$ targets inputs with $\geq 50\%$ failure probability, while lower values enforce a stricter high-probability requirement). Meanwhile, the variance term $k_t(x, x)$ drives selection toward unexplored areas within this region.

**Strategy 2: Generative synthesis (SS-Gen).** SS may overlook failure cases outside of $D_{\text{pool}}$. We extend SS to support generative failure discovery. This strategy selects $m$ "anchor" inputs from $D_{\text{pool}}$ with the highest $\alpha_{\text{SS}}$ values and uses them as in-context examples for an LLM generator: *"These test cases likely cause the target model to fail. Analyze their common features and generate a new, more challenging test case."*

**Strategy 3: Topic-aware exploration (TSS).** A limitation of SS-Gen is that the generated inputs often semantically mimic the anchors (e.g., if anchors are math problems about "counting apple", the LLM generates more "counting apple" problems). To force diversity, TSS decouples the semantic topic from the failure pattern. We partition the input space into topics $S = \{s_i\}_{i=1}^{N_{\text{topics}}}$ using BERTopic (Grootendorst, 2022) or a pre-defined set, where each topic is represented as a set of keywords (e.g., {*child, rock climbing*}, {*age, old man, counting*}).

We treat topics as arms in multi-armed bandit and select a target topic $s_t$ using UCB1 (Auer et al., 2002) to balance the reward of finding a failure case and exploration. Crucially, this selection is independent of the anchors. We take the likely-to-fail anchors (which may belong to any topic) and instruct the LLM to transpose their failure patterns into the new target topic, by adding *"Ensure the new test case belongs to this topic..."* to the SS-Gen instruction. Algorithm 2 presents the pseudocode for TSS. The detailed LLM instructions for SS-Gen and TSS are in §D.1.4 and §D.1.5.

---

**Algorithm 2** PROEVAL for Failure Case Discovery (TSS)

**Input:** Unlabeled $D_{\text{pool}}$, historical data $\mathcal{D}$ in Assumption 1, initial observations $D_0$ on the target model
**Initialize:** Cluster $D_{\text{pool}}$ into topics $S$; train GP prior on $\mathcal{D}$; get GP posterior conditioned on $D_0$.
**for** $t = 0, \ldots, T - 1$ **do**
    Select topic $s_t \in S$ via UCB1
    Select anchors $\{x_a\} \subset D_{\text{pool}}$ maximizing $\alpha_{\text{SS}}$ (Eq. 10)
    Generate $x_{t+1} \leftarrow LLM(\text{anchors} = \{x_a\}, \text{topic} = s_t)$
    $y_{t+1} \leftarrow \text{EVALUATE}(f, x_{t+1})$
    $D_{t+1} \leftarrow D_t \cup \{(x_{t+1}, y_{t+1})\}$
    Update GP posterior and topic statistics
**end for**

---

## 3. Empirical Analyses and Results

We present experimental setups (§3.1) followed by results on performance estimation (§3.2) and failure discovery (§3.3).

### 3.1. Experiment Setup

We introduce the benchmarks, models, and specific metrics used to assess PROEVAL. Additional details are provided in §C. We evaluate PROEVAL across three scenarios:

- *Default*: Predicting the performance of a known model on a new target benchmark by leveraging its historical data on other benchmarks alongside the performance of other models on that target benchmark.

- *New Model (NM)*: Evaluating a novel model with no prior performance data.

- *New Bench (NB)*: Evaluating a novel benchmark for which no prior model results are available.

We implement three types of features for the GP framework:

- *Score Features (SF)*: Applicable to the Default and New Model scenarios. In the Default setting, we employ the GMM selection approach described in §2.2.3 to curate source data. For the New Model setting, all available historical data are used.

- *Raw Prompt Features (RPF)*: This setup bypasses the learnable MLP shown in Figure 3 and defines $\psi_\theta(x)$ directly as the embedding of $x$ from the fixed embedding model. In BQ estimation, the GP prior mean is derived from a GMM-selected subset for the Default scenario, all auxiliary models for New Model, and a constant 0.5 for New Bench.

- *Tuned Prompt Features (TPF)*: This setup optimizes the MLP in Figure 3 using the objective function detailed in §2.2.2. The mean function is $\hat{\mu}(x) = \frac{1}{d}\psi_\theta(x) \times 1_d$.

Both RPF and TPF setups use kernel function $\hat{k}(x, x') = k_{\text{Matérn}}(\phi_\theta(x), \phi_\theta(x'))$, with $\phi_\theta$ defined in Equation (8).

### 3.1.1. DATASETS AND MODELS

**Datasets.** We evaluate PROEVAL across three core domains: reasoning, general world knowledge, and safety alignment with both text and image modalities, including GSM8K (Cobbe et al., 2021) and SVAMP (Patel et al., 2021) for math, StrategyQA (Geva et al., 2021) for implicit reasoning, and GQA (Hudson & Manning, 2019) for visual reasoning; MMLU (Hendrycks et al., 2021) (Professional Law subsets); ToxicChat (Lin et al., 2023), Google Civil Comments (Jigsaw) (Borkan et al., 2019), DICES-350 (Aroyo et al.,

2023) and text-to-image DIVE (Rastogi et al., 2025), where models act as reward models, and we measure alignment with human safety labels. More details in §C.1.

**Models.** We evaluate 16 LLMs and VLMs, assigning symbols for brevity. *Google (G):* Gemma-3-12B (G0) (Team et al., 2025), Gemma-3-27B (G1) (Team et al., 2025), Gemini 2.5 Flash (G2) and Pro (G3) (Gemini Team Google, 2024; Comanici et al., 2025), and Gemini 3 Flash (G4) and Pro (G5) (Team & DeepMind, 2025). *OpenAI (O):* GPT-3.5 Turbo (O1) (OpenAI, 2024), GPT-4o (O2) (Hurst et al., 2024), and GPT-5 (O3), 5.1 (O4), 5.2 (O5) (OpenAI, 2025a;b;c). *Anthropic (C):* Claude 3.5 Haiku (C1) (Anthropic, 2024), Claude 3.7 Sonnet (C2) (Anthropic, 2025a), and Claude 4.5 Sonnet (C3) and Opus (C4) (Anthropic, 2025c;b). *Qwen (Q):* Qwen3-32B (Q1) (Yang et al., 2025). For multi-modal benchmarks, we exclude architectures lacking visual support.

The performance score $y_{ij}$ for these models is 1 for failure (incorrect answer or misalignment) and 0 for success. We use the `text-embedding-3-large` embedding model from OpenAI as the fixed embedding in the GP encoder (Figure 3) and for computing embedding diversity (§3.1.2).

### 3.1.2. EVALUATION METRICS

**Performance Estimation.** To evaluate the quality and efficiency of our performance estimation method, we rely on two primary metrics. For quality, we measure the *Mean Absolute Error (MAE)*, defined as the absolute difference between the estimated average error $\hat{S}_t = \frac{1}{M}\sum_{j=1}^{M}\hat{\mu}_t(x_j)$ and the ground truth estimate $S^* = \frac{1}{M}\sum_{j=1}^{M}f(x_j)$, computed as MAE $= |\hat{S}_t - S^*|$. For efficiency, we measure the *Number of Samples (@1% MAE)*, which tracks the minimum number of samples required to achieve an MAE of $\leq 1\%$.

**Failure Discovery.** To evaluate PROEVAL's ability to uncover model vulnerabilities, we categorize our metrics into two dimensions:

- **Quality and Efficiency Metrics:** We track *Cumulative Failures* (the total number of failures discovered) alongside the *Failure Rate (FR)* (the percentage of evaluated inputs that successfully trigger a failure). Efficiency is captured by the *Samples to First Failure (SFF)*, denoting the number of queries required to identify the initial failure case.

- **Diversity Metrics:** We quantify the variety of discovered failures across both feature and semantic spaces. *Embedding Diversity* uses the normalized log-determinant of the embedding Gram matrix (Kulesza et al., 2012): $D_{emb} = \frac{1}{n}\log\det(K + \epsilon I)$, where $K_{ij} = e_i^\top e_j$ for L2-normalized embeddings and $\epsilon =$

$10^{-6}$. This measures the volume spanned in the feature space. We fix $n = 100$ samples to ensure a fair comparison and normalize to $[0, 1]$. *Topic Entropy* is measured using the Shannon entropy of the topic distribution, normalized by the maximum possible entropy: $H_{norm} = \frac{-\sum_{t \in T} p(t) \log_2 p(t)}{\log_2 |T|}$, where $p(t)$ is the proportion of samples in topic $t$, and $|T|$ is the number of unique topics. Higher entropy (up to 100%) indicates more balanced topic coverage. Finally, *Overall Diversity* summarizes both semantic and topical variety via a composite score: Diversity $= w_1 \cdot H_{norm} + w_2 \cdot \min\left(\frac{D_{emb}}{2}, 1\right)$, where $w_1 = w_2 = 0.5$ by default.

Intuitively, this composite diversity score penalizes near-duplicates and mode collapse. For example, an overall score around 0.5 indicates that, while discovered failures may span several topics, they remain semantically similar, reflecting minor variations of the same failure pattern rather than truly distinct vulnerabilities. Conversely, a high overall diversity score (e.g., $\geq 0.90$) indicates the discovery of highly distinct, semantically unique vulnerabilities spread evenly across multiple topics.

### 3.2. Results on Performance Estimation

We compare PROEVAL against *random sampling* and four *active testing* strategies (Kossen et al., 2021). The active baselines employ surrogate models, Logistic Regression (LR) or Random Forest (RF), to guide sampling, corrected by either standard Importance Sampling (IS) or LURE weighting (details can be found in Section C.1.1). We compare two estimation variants: BQ, our standard approach using transfer learning and the posterior mean sum, and BQ ROUNDED, which exploits the binary nature of the scores by rounding the posterior estimates to $\{0, 1\}$ before summation[2]. We also study the effectiveness of the active selection approach in §2.3 by comparing it to random selection. Table 1 shows the MAE at a budget of 1% of the benchmark size across these settings. Comparing BASELINES and ACTIVE SELECTION + BQ in Table 1, PROEVAL variants consistently outperform all baselines. Comparing ACTIVE SELECTION + BQ and RANDOM SELECTION + BQ, our active selection approach for BQ-SF and BQ-TPF is almost always better than random selection, highlighting the usefulness of considering the variance reduction in BQ.

Figure 10 further illustrates the convergence behavior by plotting the per-step MAE over 20 acquisition steps on four representative benchmarks. BQ-TPF and BQ-RPF start with

---

[2]While this leverages the discrete nature of the task, it discards the nuanced uncertainty information provided by the GP posterior. As a result, BQ Rounded can introduce noise and exhibits less stability than the standard BQ variant, especially when the posterior mean is near 0.5.

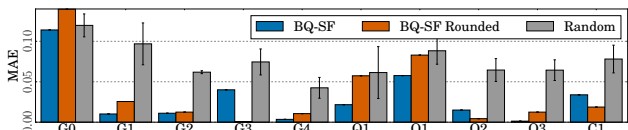

*Figure 4.* Performance estimation MAE on StrategyQA using a 1% sampling budget across 16 target models. Bayesian Quadrature (BQ and BQ Rounded) generally achieves lower estimation error compared to Random Sampling. Model symbols (e.g., G1, O1, C1) correspond to the definitions in §3. G0 failures due to weak partnership with other models (check Figure 11 for details).

higher initial error but steadily improve as more samples are acquired, with BQ-TPF generally converging faster due to the tuned prompt encoder capturing finer-grained semantic similarities. BQ-SF, benefiting from the transferred prior, achieves low error almost immediately; Figure 5 quantifies this, showing that it often requires only 1–2 evaluations to reach 1% estimation error. This efficiency stems from three factors: (1) the strong prior from transfer learning, which skips the cold-start phase; (2) BQ's ability to produce accurate aggregate estimates even with pointwise uncertainty; and (3) the active selection of inputs that maximize information gain for the integral estimate.

In the *New Bench (NB)* scenario, which falls under the prompt feature transfer framework (§2.2.2), success depends entirely on learned semantic similarities between prompts. As shown in Table 1, although performance drops relative to the Default setting, BQ-RPF still outperforms *Random Sampling* on 6 out of 9 datasets (falling behind only on DICES, StrategyQA, and SVAMP), demonstrating strong generalization in this challenging zero-shot transfer task.

We expand this analysis to all target models in Figure 4 with StrategyQA as the target benchmark. Note that if we perform GMM clustering on SVAMP (Figure 8) to select the source data, G0, O1 and Q1 would be flagged as outliers, because they are not in the same GMM cluster as the other models. When these outlier models serve as targets, their behavior deviates significantly from the source data, making it difficult to learn an informative prior. Despite this, PROEVAL's performance remains competitive. More results on other target models can be found in §C.3.1. We study negative transfer in §C.3.

**Ablation on thinking trace embeddings.** We investigate whether incorporating the target model's Chain-of-Thought (CoT) (Wei et al., 2022) trace improves BQ estimation. We compare a *Question-Only (Q-Only)* baseline against two integration strategies: *Input Concatenation (Concat)*, which appends trace text to the query, and *Latent Fusion (Fusion)*, which concatenates separate query and trace embeddings. No pre-training is done in this ablation. Table 2 shows that while reasoning traces generally reduce error compared to *Q-Only*, the effective integration method varies by task.

| METHOD | DICES | DIVE | GQA | GSM8K | JIGSAW | MMLU | STRATEGYQA | SVAMP | TOXICCHAT |
|---|---|---|---|---|---|---|---|---|---|
| *Baselines* | | | | | | | | | |
| RANDOM SAMPLING | 0.100±0.042 | 0.108±0.078 | 0.052±0.042 | 0.040±0.005 | 0.053±0.049 | 0.063±0.050 | 0.062±0.004 | 0.065±0.029 | 0.064±0.044 |
| RF+IS | 0.079±0.081 | 0.033±0.022 | 0.088±0.058 | 0.059±0.032 | 0.088±0.076 | 0.122±0.080 | 0.062±0.043 | 0.041±0.000 | 0.066±0.045 |
| LR+IS | 0.086±0.050 | 0.082±0.042 | 0.052±0.052 | 0.060±0.034 | 0.094±0.080 | 0.127±0.089 | 0.059±0.047 | 0.094±0.079 | 0.034±0.027 |
| RF+LURE | 0.080±0.085 | 0.129±0.073 | 0.108±0.050 | 0.034±0.007 | 0.103±0.046 | 0.092±0.078 | 0.028±0.029 | 0.082±0.081 | 0.066±0.023 |
| LR+LURE | 0.149±0.051 | 0.083±0.047 | 0.089±0.067 | 0.054±0.023 | 0.120±0.049 | 0.072±0.053 | 0.092±0.066 | 0.053±0.024 | 0.021±0.017 |
| *Random Selection + BQ* | | | | | | | | | |
| BQ-RPF RAND | 0.105±0.073 | 0.078±0.051 | 0.091±0.078 | 0.081±0.039 | 0.069±0.058 | 0.073±0.053 | 0.139±0.087 | 0.110±0.035 | 0.082±0.035 |
| BQ-TPF RAND | 0.078±0.057 | 0.091±0.081 | 0.041±0.014 | 0.048±0.030 | 0.063±0.039 | 0.057±0.020 | 0.064±0.055 | 0.020±0.000 | 0.038±0.039 |
| BQ-SF RAND | 0.064±0.029 | 0.020±0.030 | 0.023±0.027 | 0.023±0.016 | 0.033±0.022 | 0.048±0.019 | 0.015±0.004 | 0.015±0.003 | **0.001±0.001** |
| *Active Selection + BQ* | | | | | | | | | |
| BQ-RPF | 0.178 | 0.117 | 0.177 | 0.018 | 0.048 | 0.080 | 0.128 | 0.083 | 0.125 |
| BQ-RPF ROUNDED | 0.233 | 0.273 | 0.280 | 0.042 | 0.127 | 0.081 | 0.067 | 0.041 | 0.033 |
| BQ-TPF | 0.253 | 0.034 | 0.079 | 0.044 | 0.033 | 0.073 | 0.022 | 0.017 | 0.099 |
| BQ-TPF ROUNDED | 0.189 | 0.129 | 0.100 | 0.042 | 0.029 | 0.225 | 0.046 | 0.041 | 0.054 |
| BQ-SF | 0.067 | **0.002** | 0.016 | **0.013** | 0.016 | **0.006** | **0.011** | 0.005 | 0.002 |
| BQ-SF ROUNDED | 0.064 | 0.065 | **0.013** | 0.015 | **0.001** | 0.083 | 0.012 | 0.009 | **0.001** |
| *Special Scenarios* | | | | | | | | | |
| BQ-RPF (NB, NM) | 0.260 | 0.031 | 0.016 | 0.017 | 0.039 | 0.053 | 0.134 | 0.083 | 0.003 |
| BQ-RPF ROUNDED (NB, NM) | 0.267 | 0.050 | 0.084 | 0.042 | 0.118 | 0.205 | 0.182 | 0.041 | 0.054 |
| BQ-TPF (NB) | 0.252 | 0.037 | 0.080 | 0.045 | 0.032 | 0.073 | 0.022 | 0.014 | 0.099 |
| BQ-TPF ROUNDED (NB) | 0.164 | 0.162 | 0.109 | 0.042 | 0.030 | 0.225 | 0.045 | 0.041 | 0.054 |
| BQ-TPF (NM) | 0.105 | 0.040 | 0.171 | 0.021 | 0.006 | 0.142 | 0.016 | 0.008 | 0.037 |
| BQ-TPF ROUNDED (NM) | 0.121 | 0.287 | 0.302 | 0.042 | 0.079 | 0.225 | 0.090 | 0.041 | 0.054 |
| BQ-SF (NM) | **0.038** | 0.048 | 0.016 | 0.017 | 0.028 | 0.007 | **0.011** | **0.004** | 0.003 |
| BQ-SF ROUNDED (NM) | 0.056 | 0.055 | **0.013** | 0.015 | 0.031 | 0.083 | 0.022 | 0.009 | 0.003 |

*Table 1.* Mean Absolute Error (MAE, ↓) for **Gemini 2.5 Flash** at a 1% labeling budget. Methods use either Raw Prompt Features (RPF), Tuned Prompt Features (TPF), or Score Features (SF) as defined in §3.1. The bottom block evaluates stricter generalization under the **New Bench. (NB)** and **New Model. (NM)** scenarios. Best result per benchmark is **bolded and colored green**; values with ± report standard deviation over 5 runs. ProEval variants, especially BQ-SF and BQ-SF ROUNDED performed especially well in the Default and NM scenarios, highlighting the effectiveness of both transfer learning and active selection.

| Method | StrategyQA | GSM8K | SVAMP | JigSaw |
|---|---|---|---|---|
| Q-Only | 0.128 | 0.018 | 0.083 | 0.048 |
| Concat | 0.099 | 0.022 | 0.213 | 0.081 |
| Fusion | **0.071** | **0.009** | **0.063** | **0.029** |

*Table 2.* Ablation on reasoning trace strategies for BQ-RPF performance estimation of **Gemini 2.5 Flash**. *Q-Only* embeds only the question text, *Concat* concatenates the question with the model's reasoning trace, and *Fusion* computes a weighted average of question and reasoning embeddings ($\alpha=0.7$). All methods use a Matérn kernel with PCA-reduced embeddings (16D) and a neutral prior. We report MAE at a 1% labeling budget (lower is better).

### 3.3. Results on Failure Case Discovery

We evaluate failure case discovery on two distinct reasoning tasks: implicit reasoning (mimicking StrategyQA) and mathematical reasoning (mimicking GSM8K). We employ Gemini 2.5 Flash as the target model (deterministic decoding, temperature $\tau = 0$) and Gemini 3 Pro as the query generator (temperature $\tau = 0.7$). All experiments are conducted over 10 runs with a budget of $T = 100$ iterations.

We compare PROEVAL against baselines in two settings. For sampling from the static $D_{\text{pool}}$, we use random sampling (Rand). For synthesis baselines, we use *Rand-Gen*, which synthesizes inputs without guidance; *Rand-T-Gen*, which injects random topic constraints but lacks anchor problems;

and *Rand-Anchor-Gen*, which uses the same prompt structure as SS-Gen but replaces BQ-selected anchors with uniformly random ones, isolating the effect of anchor selection quality. Full prompt details are provided in §D.

Figure 6 shows that our generative strategies (SS-Gen, TSS) maintain near linear growth in cumulative failures, consistently outperforming the random baselines which exhibit flatter, slower growth. This gap is most distinct in the math domain (Figure 6c), where active methods discover significantly more failures than random generation. Additionally, the Samples to First Failure (SFF) in Table 3 shows that active strategies identify the first failure significantly faster—requiring less than 8 samples for math problems, compared to 11–27 samples for random methods.

Regarding the quality of these generated problems, Table 3 reports that TSS-RPF achieves highly competitive overall diversity scores (0.74 on StrategyQA and 0.73 on GSM8K), matching or nearing the unconstrained Rand-Anchor-Gen baseline while outperforming all other structured alternatives. This demonstrates its ability to generate diverse failures rather than collapsing on a single mode. Section C.4 includes representative generated examples.

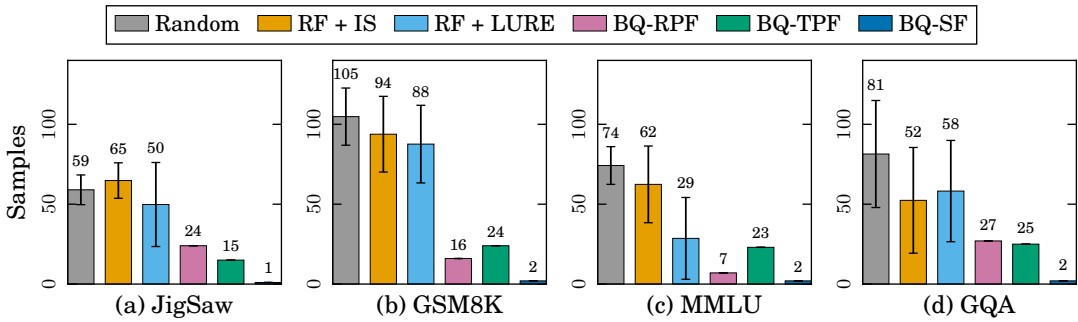

*Figure 5.* Evaluation of sampling efficiency for **Gemini 2.5 Flash**. We report the number of samples required to reach 1% MAE threshold across four benchmarks: (a) JigSaw (b) GSM8K (c) MMLU and (d) GQA. BQ achieves 1% MAE in a few samples while baselines (RF+LURE, RF+IS, Random) require 8–65× more samples.

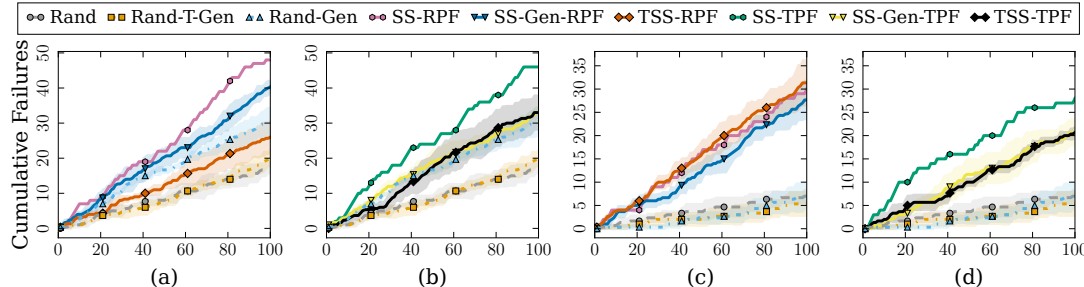

*Figure 6.* Comparison of failure discovery rates for **Gemini 2.5 Flash** (target model) using **Gemini 3 Pro** as the query generator over 100 iterations per run, averaged across 10 independent runs. Cumulative failure count for synthesized queries in: (a) Implicit Reasoning (StrategyQA) using Raw Prompt Features (RPF), (b) Implicit Reasoning (StrategyQA) using Tuned Prompt Features (TPF), (c) GSM8K-like Math problems using Raw Prompt Features (RPF), and (d) GSM8K-like Math problems using Tuned Prompt Features (TPF). Shaded regions denote standard deviation across runs.

| Method | StrategyQA | | | | | GSM8K | | | | |
|---|---|---|---|---|---|---|---|---|---|---|
| | Topic Entropy | Emb. Diversity | Overall Diversity | Failure Rate | SFF | Topic Entropy | Emb. Diversity | Overall Diversity | Failure Rate | SFF |
| SS-RPF | 71.9% | 0.71 | 0.54 | **48.0**% | 3.0 | 75.8% | 0.70 | 0.55 | 30.0% | 4.0 |
| SS-TPF | 65.0% | 0.75 | 0.51 | 46.0% | 4.0 | 78.7% | 0.73 | 0.58 | 28.0% | 4.0 |
| Rand | 68.4% | **1.00** | 0.59 | 17.3% | 9.3 | 67.1% | **1.00** | 0.59 | 7.0% | 11.7 |
| SS-Gen-RPF | 95.2% | 0.95 | 0.71 | 40.3% | 3.0 | 96.2% | 0.99 | 0.73 | 27.7% | 4.3 |
| SS-Gen-TPF | 96.8% | 0.97 | 0.73 | 33.0% | **2.0** | 96.8% | 0.96 | 0.72 | 20.7% | 7.7 |
| TSS-RPF | 99.0% | **1.00** | **0.74** | 26.0% | 4.3 | 98.8% | 0.95 | 0.73 | **31.3**% | **3.3** |
| TSS-TPF | 96.5% | 0.98 | 0.73 | 33.0% | 4.7 | 98.1% | 0.94 | 0.73 | 20.7% | 6.7 |
| Rand-Anchor-Gen | 99.5% | 0.98 | 0.74 | 22.3% | 4.3 | 99.0% | **1.00** | **0.75** | 6.3% | 27.3 |
| Rand-T-Gen | 99.2% | 0.87 | 0.71 | 20.0% | 4.3 | 98.6% | 0.95 | 0.73 | 5.3% | 17.3 |
| Rand-Gen | **99.3**% | 0.57 | 0.64 | 30.3% | 3.7 | **99.6**% | 0.78 | 0.69 | 7.3% | 27.0 |

*Table 3.* Failure discovery performance on StrategyQA and GSM8K. We evaluate Gemini 2.5 Flash as the target model with Gemini 3 Pro as the generator, over 100 iterations per run. **Bold** values indicate the best performance in each column per dataset.

# 4. Conclusion

We introduced PROEVAL, a proactive evaluation framework that uses Bayesian ideas and transfer learning to improve the sample efficiency and effectiveness of both performance estimation and failure case discovery. This is especially important for expensive-to-query and expensive-to-rate modern generative AI models. Our theoretical and empirical studies show strong promise of our proposed approach. In particular, ProEval achieves a 8-65x reduction on sample sizes for evaluation, and discovers 2-5x more failure cases than competitive baselines. Future work includes reducing the reliance on high-quality embeddings, developing better acquisition functions, surrogate models, variance reduction techniques, and strategies to take into account the different rater costs and costs for generating model responses based on different inputs.

## Impact Statement

PROEVAL is a framework for efficient evaluation and failure detection of resource-intensive generative AI models. By significantly reducing evaluation overhead, our approach accelerates GenAI research and rapid quality iteration. Furthermore, systematically uncovering and forecasting model failures deepens our fundamental understanding of these systems, laying the groundwork for more trustworthy AI. Ultimately, we anticipate several positive societal outcomes: enhanced transparency regarding model limitations, reduced energy consumption through optimized testing, and the development of safer, more equitable models by prioritizing diverse and challenging test cases.

## Acknowledgments

We would like to thank Zoubin Ghahramani, Virginia Aglietti, Mani Malek, Neha Kalibhat, Been Kim, Noah Fiedel, Jason Baldridge, Aditya Mone, Manoj Middepogu, Oyvind Tafjord and others for discussions on Bayesian quadrature, active testing, automated red teaming, GenAI evaluation and/or contributions to early versions of this work. We would also like to thank David Madras, Tamara Broderick's group and anonymous reviewers for detailed and insightful feedback.

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

# A. Literature Review

**Efficient evaluation.** The challenge of evaluating generative models under minimal budgets has led to two dominant research directions: static benchmark pruning, which seeks to identify a fixed, representative subset of queries, and standard active testing, which dynamically selects queries during evaluation. ProEval fundamentally differs from both in its underlying assumptions and statistical implications.

- **Static benchmark pruning.** Methods such as *TinyBenchmarks* (Polo et al., 2024) and *MetaBench* (Kipnis et al., 2025) aim to identify a fixed, representative subset of examples that approximate global performance. Similarly, Perlitz et al. (2024) analyze the reliability-compute tradeoff in benchmarks like HELM, proposing strategies that allocate fewer samples to lower-tier models. *Anchor Points* (Vivek et al., 2024) and DISCO (Rubinstein et al., 2025) advance this by using "source models" to identify dataset medoids or maximize inter-model disagreement. While effective for standardizing leaderboards, these approaches are inherently target-agnostic: they assume the failure modes of the target model perfectly align with the source models used to select the subset.

  If the target model exhibits idiosyncratic failures (e.g., due to a different architecture or safety tuning), static subsets fail to capture them. Recent approaches like *Fluid Benchmarking* (Hofmann et al., 2025) attempt to overcome this by combining IRT priors with computerized adaptive testing to dynamically select items based on a model's latent ability. While effective at reducing evaluation variance, this approach is restricted to static item pools—meaning it cannot synthesize novel test cases—and reduces a model's complex, multi-dimensional capabilities to a single scalar ability score. PROEVAL avoids this rigidity. While we use source models to initialize a prior, our Bayesian acquisition function allows the evaluation to deviate from the prior, synthesize new questions and explore target-specific weaknesses.

- **Dynamic active testing.** Active testing seeks to optimize sample efficiency by iteratively selecting test points. Early work by Nguyen et al. (2018) studied a special case of active testing focused on human vetting of noisy labels. Subsequent work adapted this framework to efficient model evaluation. Kossen et al. (2021; 2022) introduced unbiased importance sampling estimators (LURE) guided by surrogates trained on the target model's training data. In the pairwise setting, *DiffUse* (Ashury Tahan et al., 2024) adopts a similar active approach, clustering output difference vectors to select a diverse set of preference queries.

  However, these approaches remain limited: white-box methods like Huang et al. (2025) may fail to detect errors where the model is confidently wrong, while proxy-model-based methods might be susceptible to containing shared blindspots between the proxy and surrogate models. Also, since they have no knowledge of historical failure patterns across models, they must spend their initial budget identifying difficulty regions that are often already known to the community.

  To address this, Li et al. (2025) proposed transferring inter-prompt patterns from historical models using neural processes. However, they frame evaluation as an *imputation* problem, using reinforcement learning to predict every missing score.

  Park et al. (2025) adaptively weight human and LLM-judge labels to improve estimation, but evaluate all inputs rather than selecting which to test. Wang et al. (2025) select test points via variance-reducing partitions, but cold-start each evaluation without leveraging cross-model structure. In contrast, PROEVAL frames evaluation as *Bayesian quadrature* (integration) and uses transfer learning. We select samples to directly estimate the final aggregate metric, minimizing the error without needing to reconstruct individual data points. Furthermore, ProEval moves beyond simple estimation to perform *failure case discovery* and *query synthesis*, capabilities absent in their framework.

**Active learning.** While Active Learning (AL) shares the goal of efficient data selection, its objective is fundamentally different from ours. Standard AL methods (Atlas et al., 1989; Settles, 2009; Houlsby et al., 2011; Sener & Savarese, 2017) focus on selecting samples to *train* or *fine-tune* a model to minimize generalization error. In contrast, the model is fixed in our problem setting, and we aim to select samples to *estimate* a test metric (integration) and/or discover failure regions (level set estimation). Similarly, while early NLP methods use Bayesian Optimization to curate training data (Ruder & Plank, 2017), the model is fixed in our problem setting. For active sampling, the most relevant work to ours is (Wang et al., 2018a), and they also used Gaussian processes for superlevel set sampling, but the goal is to enable better actions in planning tasks. We aim to select samples dynamically to *estimate* a test metric and discover failure regions. Thus, our acquisition functions prioritize variance reduction of the integral estimator and/or superlevel set locations rather than information gain for model parameters.

**Simulated users and Agentic evaluation.** To capture capabilities beyond static QA, researchers increasingly rely on "simulated users" (Philipov et al., 2024) and interactive agent frameworks (Li et al., 2024; Zhao et al., 2025; Liu et al., 2023; Chang et al., 2024). Some approaches also employ diverse personas (Chen et al., 2025; Lu et al., 2025) to model real-world usage variance. However, the heavy computational cost of these simulations makes them unsuitable for iterative development. By treating the simulation as a costly oracle $f(x)$, PROEVAL uses Bayesian Active Learning to intelligently sample only the most critical test cases, allowing researchers to leverage high-fidelity evaluation at a fraction of the cost.

**Failure discovery and Red teaming.** Identifying blindspots in model capabilities is traditionally the domain of adversarial attacks and automated red teaming. Early approaches fine-tuned an "attacker" (Perez et al., 2022) or use iterative refinement strategies like *PAIR* (Chao et al., 2024) and *TAP* (Mehrotra et al., 2024) which often rely on human-specified harmful behaviors. Prompt-optimization methods (Zhou et al., 2023) can be prone to a lack of diversity, leaving vast regions of the failure landscape unexplored.

Existing solutions trade efficiency for diversity. *Rainbow Teaming* (Samvelyan et al., 2024) addresses diversity issues via evolutionary Quality-Diversity but is highly sample-inefficient. (Kalibhat et al., 2026), on the other hand, uses concept edits to effectively explore the prompt space and the size of prompts may grow exponentially. Conversely, *Bayesian Red Teaming (BRT)* (Lee et al., 2023) improves efficiency using Gaussian Processes but remains an optimization method, relying on heuristic penalties such as Self-BLEU to artificially force diversity.

Complementary to input-space searches, *white-box* methods leverage internal model states to detect failure. Recent work predicts errors by probing residual streams (Sun et al., 2025) or training confidence networks on intermediate features (Corbière et al., 2019), while others actively correct hallucinations by shifting activations along "truth directions" (Li et al., 2023).

PROEVAL fundamentally differs from these methods by treating evaluation as a *black-box* Bayesian Level Set Estimation (LSE) problem (Gotovos et al., 2013) over the input space. Rather than optimizing for a single failure mode, PROEVAL seeks to efficiently map the entire *superlevel set* of inputs where the model fails, providing a global view of performance holes with minimal sample cost.

**Bayesian quadrature and Gaussian processes.** Most active testing strategies rely on Importance Sampling (IS) (Kossen et al., 2021; Huang et al., 2025), which often suffers from high variance in high-dimensional embedding spaces. We instead formulate evaluation as Bayesian Quadrature (BQ) (O'Hagan, 1991; Ghahramani & Rasmussen, 2002).

Osborne et al. (2012) established the framework for *Active* BQ, with subsequent work optimizing for speed (Gunter et al., 2014) and parallelization (Wagstaff et al., 2018). Crucially, this body of work focuses on scientific computing tasks, such as estimating marginal likelihoods. We differentiate ourselves by adapting BQ to the distinct regime of *generative AI evaluation*, where the goal is to estimate performance metrics (e.g., accuracy, safety) over a high-dimensional semantic input space. In this setting, standard BQ suffers from a "cold start" problem, as generic priors struggle to model the complex failure landscape of LLMs. PROEVAL overcomes this by drawing on the existing knowledge of pre-trained Gaussian processes (Wang et al., 2024) and score-space transfer learning (Wang et al., 2018b; Kim et al., 2019) to construct informed priors from historical data.

## B. Proof for Theorem 3

The proof of Theorem 3 relies on the fact that the sample mean $\hat{\mu}_t(x)$ is unbiased and bounded by the posterior variance. The detailed proof is as follows.

*Proof.* By Theorem 5 of (Wang et al., 2024), we have

$$\mathbb{E}[\hat{\mu}_t(x)] = \mu_t(x) \tag{11}$$

$$|\hat{\mu}_t(x) - \mu_t(x)|^2 < a^2 k_{t-1} \circ \sigma^2(x) \tag{12}$$

where Equation (12) holds with probability $1 - \delta$, $a^2 = \frac{4\left(t + 1 + 2\sqrt{t \log \frac{4}{\delta}} + 2 \log \frac{4}{\delta} - 2/N\right)}{(N - t - 2)\delta}$.

Our integral estimator based on estimated GP mean and kernel is

$$\mathbb{E}[\hat{S}_t] = \frac{1}{M}\mathbb{E}[\sum_{j=1}^{M}\hat{\mu}_t(x_j)] = \frac{1}{M}\sum_{j=1}^{M}\mathbb{E}[\mu_t(x_j)] = \mathbb{E}[S \mid D_t],$$

which means that $\hat{S}_t$ is an unbiased estimator for the ground truth expectation of the integral. With probability $1 - \delta$, the estimate is bounded as

$$|\hat{S}_t - S_t| = \left|\frac{1}{M}\sum_{j=1}^{M}(\hat{\mu}_t(x_j) - \mu_t(x_j))\right| \le \frac{1}{M}\sum_{j=1}^{M}|\hat{\mu}_t(x_j) - \mu_t(x_j)| < a'\sqrt{\kappa + \sigma^2}.$$

Here, $a' = \left(\frac{4M\left(t+1+2\sqrt{t\log\frac{4M}{\delta}}+2\log\frac{4M}{\delta}-2/N\right)}{(N-t-2)\delta}\right)^{\frac{1}{2}}$ and $\kappa$ is a constant that bounds the value of the kernel function. $\square$

# C. Details on Experiments and Additional Results

In this section, we describe the details on the experiments and present additional results.

In our experiments, we used GMM-based approach for source data selection. The selection strategy uses the collection of all benchmarks except the target benchmark as the reference benchmark.

## C.1. Datasets

We evaluate PROEVAL's performance estimation (§3.2) and failure case discovery (§3.3) abilities across three core domains: mathematical reasoning, general world knowledge, and safety alignment. These benchmarks represent a spectrum of evaluation challenges, from objective numeric correctness to nuanced human judgment. Most datasets are subsampled to accelerate experimentation and reduce inference costs; see Section C.1 for dataset sizes.

**Mathematical and logical reasoning.** We use GSM8K (Cobbe et al., 2021) and SVAMP (Patel et al., 2021) to assess multi-step chain-of-thought reasoning. GSM8K provides high-quality grade school math problems requiring numeric solutions, while SVAMP introduces linguistic variations to test phrasing robustness. For implicit reasoning, we use StrategyQA (Geva et al., 2021), which requires the model to decompose "Yes/No" questions into latent reasoning steps (e.g., "Did Aristotle use a laptop?"). We also conducted the experiments with the model's multi-modal reasoning ability using the GQA (Hudson & Manning, 2019) dataset.

**General world knowledge.** We employ the MMLU (Hendrycks et al., 2021) benchmark, covering 57 diverse subjects across STEM, the humanities, and social sciences. We use two subsets a) abstract algebra and b) professional law for evaluating more general-purpose capabilities across varying difficulty levels in a multiple-choice format.

**Safety and human alignment.** To evaluate model safety and rater alignment, we use ToxicChat (Lin et al., 2023) and Google Civil Comments (Jigsaw) (Borkan et al., 2019), which provide real-world user-AI interactions and online comments annotated for toxicity. Furthermore, we include the Google DICES-350 (Aroyo et al., 2023) and the text-to-image DIVE (Rastogi et al., 2025) datasets, which contain expert safety ratings across 123 unique raters. Unlike the reasoning tasks, these safety datasets allow us to evaluate models as *reward models*. Here, the score $f(x)$ measures the alignment between model-predicted labels and ground-truth human annotations.

| Dataset | Modality | Our Size | Standard Size |
| --- | --- | ---: | ---: |
| DICES-350 | Text | 1,500 | $42k$ |
| DIVE | Text & Image | 1,500 | 38,410 |
| GQA | Text & Image | 2,000 | $22.7M$ |
| GSM8K | Text | 1,319 | 8,500 |
| Jigsaw | Text | 1,500 | 150,000 |
| MMLU | Text | 1,534 | 1,534 |
| StrategyQA | Text | 1,603 | 2,780 |
| SVAMP | Text | 700 | 1,000 |
| ToxicChat | Text | 1,500 | 10,000 |

*Table 4.* Comparison of dataset sizes and modalities. We utilize subsets of the original data to facilitate rapid experimentation and minimize inference costs; for MMLU, the *Professional Law* subset is used.

### C.1.1. BASELINES FOR PERFORMANCE ESTIMATION

**Baselines.** We benchmark our approach against five sampling strategies. *Random Sampling* provides a simple baseline by uniformly selecting test cases to estimate global accuracy via the sample mean. For active strategies, we adopt the *Active Testing* framework (Kossen et al., 2021), which uses surrogate models to guide acquisition while correcting for selection bias. We employ Logistic Regression (*LR*) and Random Forest (*RF*) surrogates trained on evaluation results of the 11 models to define acquisition probabilities $q(i)$. To unbias the estimates, we evaluate two weighting schemes: standard Importance Sampling (*IS*) using weights $w_m = 1/(n \cdot q(i_m))$, and *LURE*, which accounts for sequential sampling without replacement: $\frac{1}{M} \sum_{m=1}^{M} \left( 1 + \frac{N-M}{N-m} \left( \frac{1}{(N-m+1) \cdot q(i_m)} - 1 \right) \right) f_m$.

Combinations of these surrogates and estimators yield four active variants: *LR+IS*, *LR+LURE*, *RF+IS*, and *RF+LURE*.

## C.2. Ablation Studies with Pre-training Data Selection

The foundational assumption of our transfer learning approach is that while language models vary in capability, their failure modes are rarely orthogonal. To validate this, we analyze the pairwise correlation of model performance across benchmarks. As visualized in Figure 7, we observe uniformly positive correlations (indicated by the dark red blocks) across all model pairs on both StrategyQA and SVAMP. This confirms that hard questions tend to be hard for all models, allowing PROEVAL to effectively leverage historical data.

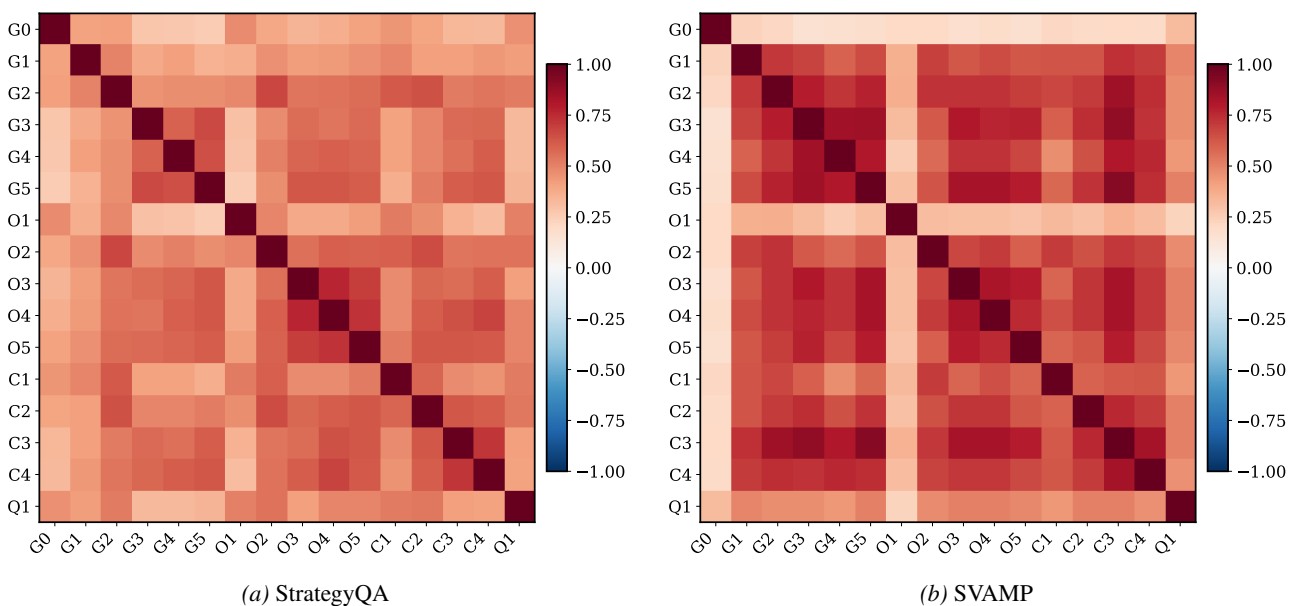

*(a)* StrategyQA        *(b)* SVAMP

*Figure 7.* Correlation between model performance on different benchmarks. Positive correlations across all model pairs (red blocks) indicate that failure modes are shared rather than idiosyncratic, validating the use of transfer learning.

Building on this shared structure, §2.2 introduces an approach to select historical datasets for pre-training the GP. Figure 8 visualizes the evaluation data from each model to show the Gaussian clusters. For example, we can use SVAMP as the hold-out benchmark to decide which models' data can be used to pre-train the GP when evaluating target model Gemini 2.5 Flash (G2) on StrategyQA or GSM8K. In this case, Gemma-3-12B (G0) and GPT-3.5 Turbo (O1) fall outside the cluster and are consequently removed from the pre-training data.

To further demonstrate that this pre-training data selection strategy is necessary, we show that blindly choosing data can significantly degrade BQ performance. Figure 9 presents an ablation study evaluating all pairwise combinations of pre-training data for two target models on GSM8K. Each cell in the heatmaps shows the final MAE when using that specific pair of models as the pre-training set, where lower values (green) indicate better prior quality. The results clearly show that pre-train selection matters significantly: poor choices of model pairs can yield up to a $100\times$ higher MAE, and the optimal pair varies by target model.

## C.3. Additional Results on Performance Estimation

**Case study on negative transfer.** To prevent negative transfer from unrelated historical data, our source data selection approach (§2.2.3) filters models by analyzing their performance clusters on a hold-out benchmark. As detailed in Section C.2, we project model profiles via PCA to identify suitable source models that share similar failure modes (visualized in Figure 8). For example, when selecting pre-training data for evaluating Gemini 2.5 Flash (G2) on GSM8K, models like G0 and O1 fall outside the cluster and are excluded. Figure 11 (corresponding to Figure 9) confirms that failing to filter these models and blindly selecting pre-training pairs can increase the estimation MAE by up to $100\times$, whereas our selection strategy successfully identifies the optimal, low-error pairs.

**Modality transfer.** We performed preliminary experiments for BQ to investigate transferring across modalities. We used DICES (text) dataset to perform the knowledge transfer, and ran PROEVAL on DIVE (image) data with 15 samples. The results show that, compared with the no knowledge transfer, the cross-modality knowledge contributes significantly,

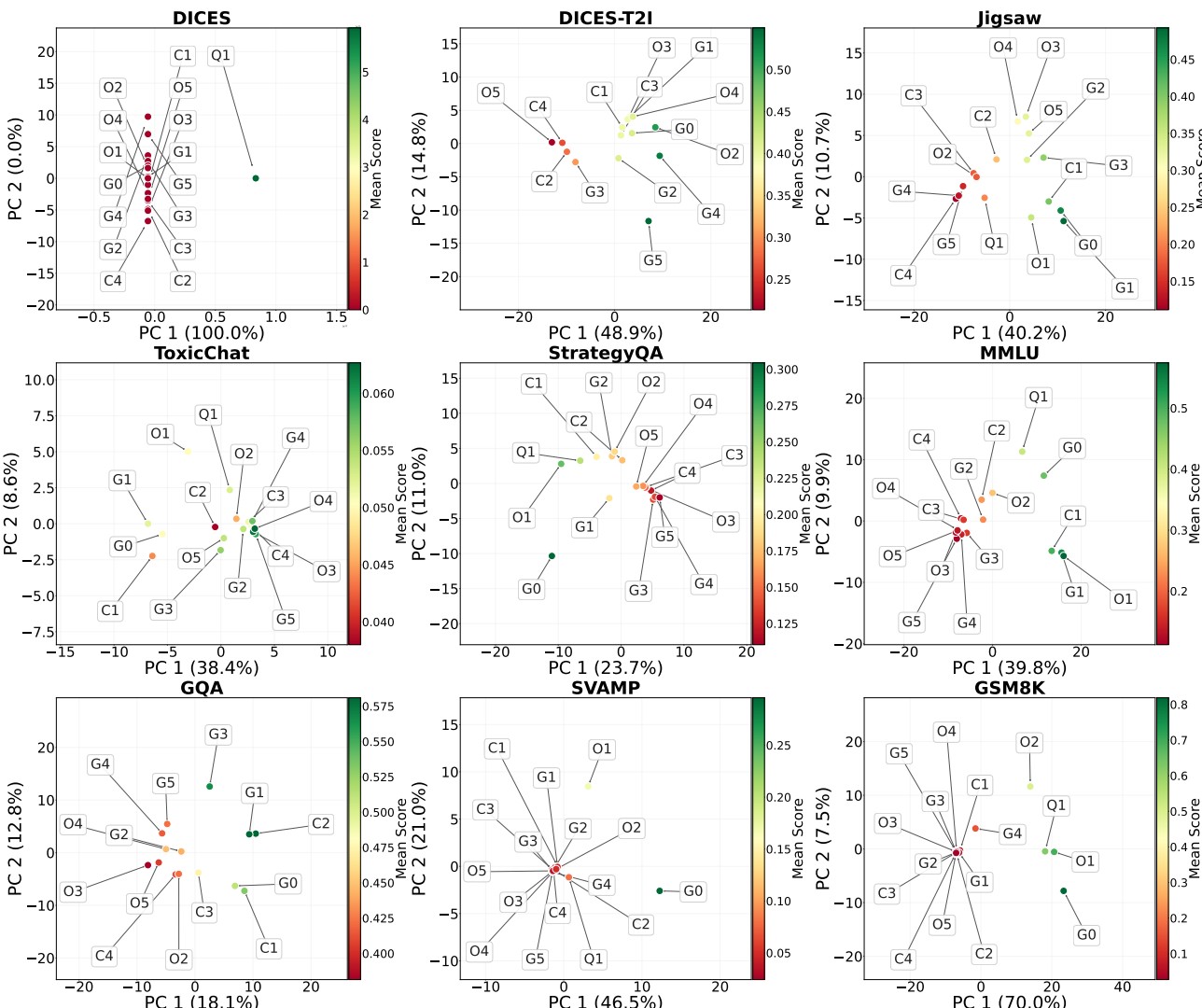

*Figure 8.* **Visualization of GenAI Model Performance Profiles across 9 Benchmarks via PCA.** Each subplot displays a 2D projection of 14 AI models based on their granular, per-question score feature matrices, derived via Principal Component Analysis (PCA) with the percentage of total explained variance indicated on the axes. The color of each marker reflects the model's average score on that specific benchmark, mapping from red (lower performance) to green (higher performance), while model names are abbreviated using short codes (e.g., G0–G5 for Google, O2–O5 for OpenAI, C1–C4 for Anthropic detailed in §3.1) to maximize legibility in dense regions. Models that cluster together make similar errors on the same questions.

reducing the MAE from 0.111 to 0.055.

**Binary data in the experiments and the Gaussian observation model.** The choice of the observation model is a deliberate trade-off to ensure computational tractability and analytical updates in BQ with pre-trained GPs. While a Bernoulli/probit link is standard for binary data, it precludes a closed-form posterior for the integral and the pre-training objective. This would require approximate inference (e.g., variational inference, MCMC), which introduces approximation errors that may defeat the purpose of accurately modeling binary data.

We experimented with a standard GP classifier (GPC) using Laplace approximation (Rasmussen & Williams, 2006) and the approximated log marginal likelihood for the pre-training. In the settings of Table 1 at a 1% budget, GPC often underperforms, yielding an MAE of 0.1653 on StrategyQA and 0.0516 on SVAMP. More investigations on transfer learning for GPC is required to make concrete claims about how to best use GPC for our tasks. The "BQ Rounded" variant's results, while varied, provide a baseline for future work in non-Gaussian BQ for GenAI evaluation.

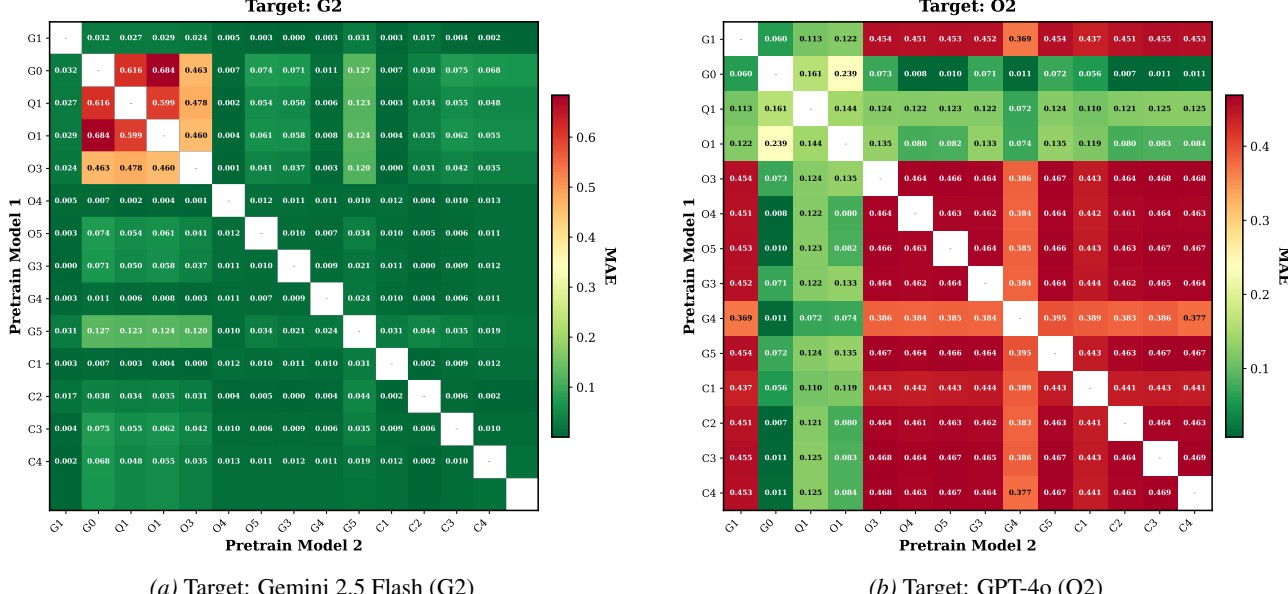

*(a)* Target: Gemini 2.5 Flash (G2)   *(b)* Target: GPT-4o (O2)

*Figure 9.* Pretrain model pair ablation on GSM8K. Each cell shows MAE when using that pair of models as the pretrain set. The results show that (1) pretrain selection matters significantly—poor pairs yield 100× higher MAE, and (2) the optimal pretrain pair varies by target model.

## C.3.1. RESULTS ON DIFFERENT TARGET MODELS

Apart from the Gemini 2.5 Flash model, we extended our evaluation of PROEVAL to a comprehensive set of other frontier and legacy generative AI models using the same experimental setting as Table 1. The full results are presented in Tables 5 through 19. As shown in these results, the *Active Selection + BQ* methods consistently achieve the lowest MAE in the majority of benchmarks, outperforming both random sampling and strong importance sampling baselines like LURE. This demonstrates that the variance-reduction property of active BQ acquisition generalizes well across different model architectures.

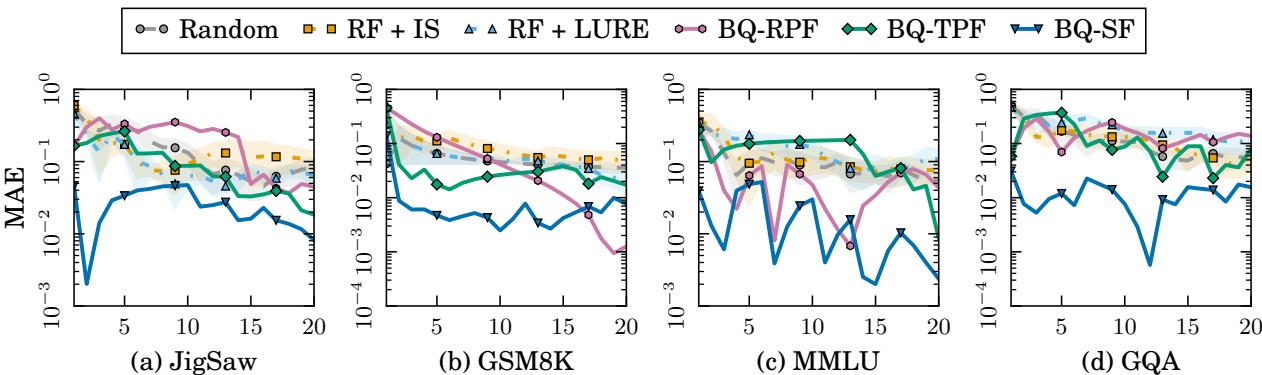

*Figure 10.* Per-step MAE across sampling iterations for estimating the error rate of **Gemini 2.5 Flash** with a budget of 20 samples. Results are shown for (a) JigSaw, (b) GSM8K, (c) MMLU, and (d) GQA. Each plot displays the instantaneous MAE at each acquisition step, averaged over 5 independent runs with standard error of the mean (SEM) shading.

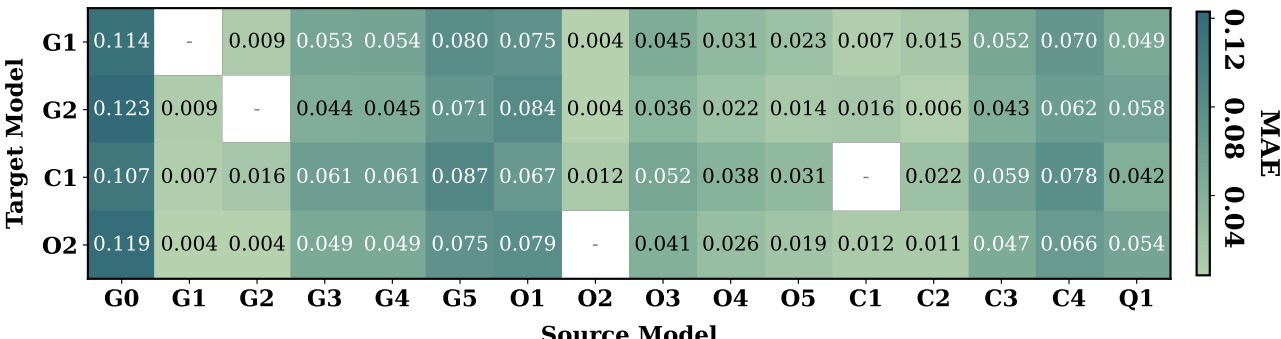

*Figure 11.* **Impact of negative transfer.** We analyze the MAE of the BQ estimator with 4 target models (G1, G2, C1, O2 on the Y-axis) when using a prior constructed by pairing the target with one of the 16 source models (X-axis). The consistently higher error rates observed when pairing with dissimilar source models (e.g., G0 with MAE >0.10, and G5, O1 with MAE >0.07) confirm that blindly including misaligned historical data harms estimation, while closely related models (e.g., G1↔G2 with MAE <0.01) enable effective transfer, validating the necessity of our Gaussianity-based filtering approach.

| METHOD | DICES | DIVE | GQA | GSM8K | JIGSAW | MMLU | STRATEGYQA | SVAMP | TOXICCHAT |
|---|---|---|---|---|---|---|---|---|---|
| RANDOM SAMPLING | 0.057±0.031 | — | — | 0.076±0.065 | 0.089±0.084 | 0.105±0.058 | 0.088±0.037 | 0.083±0.070 | 0.037±0.016 |
| RF+IS | 0.060±0.030 | — | — | 0.143±0.064 | 0.102±0.091 | 0.060±0.041 | 0.099±0.043 | 0.128±0.096 | 0.047±0.026 |
| LR+IS | 0.072±0.057 | — | — | 0.062±0.052 | 0.084±0.053 | 0.052±0.044 | 0.089±0.064 | 0.177±0.145 | 0.023±0.013 |
| RF+LURE | 0.071±0.034 | — | — | 0.106±0.091 | 0.075±0.095 | 0.103±0.104 | 0.091±0.059 | 0.121±0.069 | 0.037±0.016 |
| LR+LURE | 0.063±0.055 | — | — | 0.119±0.080 | 0.095±0.042 | 0.113±0.044 | 0.100±0.088 | 0.195±0.129 | 0.037±0.016 |
| *Random Selection + BQ* | | | | | | | | | |
| BQ-RPF RAND | 0.065±0.056 | — | — | 0.110±0.064 | 0.041±0.049 | 0.123±0.061 | 0.057±0.045 | 0.120±0.097 | 0.031±0.022 |
| BQ-TPF RAND | **0.057±0.025** | — | — | 0.040±0.019 | 0.066±0.045 | 0.120±0.082 | 0.065±0.031 | 0.032±0.049 | 0.047±0.024 |
| BQ-SF RAND | 0.108 | — | — | 0.206±0.045 | 0.085±0.028 | 0.164±0.027 | 0.083±0.017 | 0.102±0.033 | **0.002±0.001** |
| *Active Selection + BQ* | | | | | | | | | |
| BQ-RPF | 0.072 | — | — | 0.143 | 0.183 | 0.018 | 0.079 | **0.018** | 0.039 |
| BQ-RPF ROUNDED | 0.087 | — | — | 0.294 | 0.154 | **0.004** | 0.117 | 0.116 | 0.013 |
| BQ-TPF | 0.061 | — | — | **0.011** | 0.010 | 0.106 | **0.042** | 0.488 | 0.033 |
| BQ-TPF ROUNDED | 0.095 | — | — | 0.306 | 0.085 | 0.222 | 0.132 | 0.831 | 0.050 |
| BQ-SF | 0.108 | — | — | 0.116 | 0.020 | 0.137 | 0.058 | 0.122 | 0.005 |
| BQ-SF ROUNDED | 0.108 | — | — | 0.218 | **0.007** | 0.085 | 0.083 | 0.136 | 0.012 |

*Table 5.* MAE (↓) at 1% budget for **GPT-3.5 Turbo**. ± indicates mean ± std over 5 runs; active BQ is deterministic. Best results are **bolded**. The model doesn't support image inputs so we don't have DIVE and GQA numbers.

| METHOD | DICES | DIVE | GQA | GSM8K | JIGSAW | MMLU | STRATEGYQA | SVAMP | TOXICCHAT |
|---|---|---|---|---|---|---|---|---|---|
| RANDOM SAMPLING | 0.107±0.043 | 0.081±0.049 | 0.074±0.032 | 0.115±0.071 | 0.085±0.030 | 0.126±0.066 | 0.064±0.032 | 0.076±0.022 | 0.049±0.022 |
| RF+IS | 0.080±0.078 | 0.074±0.052 | 0.101±0.050 | 0.102±0.047 | 0.066±0.056 | 0.089±0.035 | 0.079±0.089 | 0.049 | 0.047±0.002 |
| LR+IS | 0.079±0.052 | 0.118±0.029 | 0.067±0.037 | 0.157±0.083 | 0.101±0.053 | 0.041±0.033 | 0.069±0.043 | 0.095±0.073 | 0.026±0.010 |
| RF+LURE | 0.072±0.051 | 0.091±0.050 | 0.098±0.063 | 0.103±0.083 | 0.083±0.057 | 0.133±0.061 | 0.077±0.026 | 0.076±0.022 | 0.043±0.012 |
| LR+LURE | 0.080±0.046 | 0.124±0.078 | 0.084±0.055 | 0.072±0.063 | 0.078±0.031 | 0.054±0.048 | 0.068±0.021 | 0.067±0.022 | 0.026±0.010 |
| *Random Selection + BQ* | | | | | | | | | |
| BQ-RPF RAND | 0.078±0.037 | 0.165±0.111 | 0.046±0.040 | 0.113±0.059 | 0.058±0.041 | 0.078±0.042 | 0.066±0.014 | 0.010±0.007 | 0.027±0.014 |
| BQ-TPF RAND | 0.031±0.025 | 0.041±0.045 | 0.066±0.023 | 0.088±0.056 | 0.039±0.029 | 0.104±0.053 | 0.048±0.037 | 0.023±0.002 | 0.028±0.012 |
| BQ-SF RAND | 0.027±0.003 | 0.070±0.012 | 0.055±0.028 | 0.057±0.033 | 0.050±0.032 | 0.040±0.030 | 0.015±0.010 | 0.013±0.009 | 0.008 |
| *Active Selection + BQ* | | | | | | | | | |
| BQ-RPF | 0.078 | 0.096 | 0.027 | 0.009 | 0.038 | **0.030** | 0.012 | 0.003 | 0.002 |
| BQ-RPF ROUNDED | 0.061 | 0.099 | 0.021 | **0.000** | 0.025 | 0.117 | 0.041 | 0.016 | **0.001** |
| BQ-TPF | 0.072 | **0.032** | 0.153 | 0.113 | 0.062 | 0.190 | 0.097 | 0.017 | 0.029 |
| BQ-TPF ROUNDED | 0.151 | 0.262 | 0.421 | 0.420 | 0.022 | 0.269 | 0.185 | 0.049 | 0.045 |
| BQ-SF | **0.022** | 0.093 | 0.026 | 0.019 | 0.051 | 0.088 | 0.015 | **0.003** | 0.008 |
| BQ-SF ROUNDED | 0.058 | 0.079 | **0.002** | 0.144 | **0.018** | 0.140 | **0.004** | 0.016 | **0.001** |

*Table 6.* MAE (↓) at 1% budget for **GPT-4o**. ± indicates mean ± std over 5 runs; active BQ is deterministic. Best results are **bolded**.

| METHOD | DICES | DIVE | GQA | GSM8K | JIGSAW | MMLU | STRATEGYQA | SVAMP | TOXICCHAT |
|---|---|---|---|---|---|---|---|---|---|
| RANDOM SAMPLING | 0.125±0.117 | 0.041±0.052 | 0.076±0.028 | 0.032±0.004 | 0.043±0.031 | 0.087±0.051 | 0.064±0.029 | 0.065±0.031 | 0.052±0.024 |
| RF+IS | 0.134±0.101 | 0.106±0.023 | 0.097±0.069 | 0.055±0.034 | 0.087±0.011 | 0.050±0.035 | 0.115±0.054 | 0.040 | 0.075±0.020 |
| LR+IS | 0.067±0.051 | 0.094±0.098 | 0.101±0.040 | 0.037±0.005 | 0.174±0.088 | 0.113±0.061 | 0.053±0.029 | 0.094±0.080 | 0.018±0.026 |
| RF+LURE | 0.180±0.095 | 0.071±0.057 | 0.086±0.080 | 0.035±0.005 | 0.131±0.059 | 0.072±0.044 | 0.024±0.013 | 0.065±0.031 | 0.037±0.030 |
| LR+LURE | 0.093±0.082 | 0.093±0.053 | 0.041±0.022 | 0.030 | 0.077±0.065 | 0.039±0.030 | 0.065±0.029 | 0.053±0.025 | 0.067±0.043 |
| *Random Selection + BQ* | | | | | | | | | |
| BQ-RPF RAND | 0.028±0.007 | 0.056±0.037 | 0.044±0.022 | 0.026±0.010 | 0.093±0.030 | 0.059±0.035 | 0.079±0.069 | 0.009±0.004 | 0.037±0.017 |
| BQ-TPF RAND | 0.038±0.032 | 0.032±0.022 | 0.109±0.063 | 0.041±0.031 | 0.121±0.104 | 0.057±0.020 | 0.052±0.022 | 0.015±0.002 | 0.043±0.023 |
| BQ-SF RAND | 0.072±0.015 | 0.022±0.015 | 0.057±0.018 | 0.032±0.010 | 0.046±0.026 | 0.065±0.025 | 0.016±0.010 | 0.016±0.007 | 0.010 |
| *Active Selection + BQ* | | | | | | | | | |
| BQ-RPF | **0.012** | 0.227 | **0.007** | **0.000** | **0.003** | 0.052 | 0.075 | **0.006** | 0.005 |
| BQ-RPF ROUNDED | 0.073 | 0.237 | 0.016 | 0.004 | 0.031 | 0.022 | 0.011 | 0.007 | 0.011 |
| BQ-TPF | 0.191 | 0.039 | 0.109 | 0.009 | 0.097 | 0.103 | 0.129 | 0.008 | 0.046 |
| BQ-TPF ROUNDED | 0.620 | 0.331 | 0.382 | 0.030 | 0.174 | 0.123 | 0.145 | 0.040 | 0.062 |
| BQ-SF | 0.023 | 0.038 | 0.051 | 0.029 | 0.013 | 0.037 | **0.001** | 0.007 | **0.005** |
| BQ-SF ROUNDED | 0.012 | **0.006** | 0.033 | 0.002 | 0.019 | **0.002** | 0.012 | 0.007 | 0.007 |

*Table 7.* MAE (↓) at 1% budget for **GPT-5**. ± indicates mean ± std over 5 runs; active BQ is deterministic. Best results are **bolded**.

| METHOD | DICES | DIVE | GQA | GSM8K | JIGSAW | MMLU | STRATEGYQA | SVAMP | TOXICCHAT |
|---|---|---|---|---|---|---|---|---|---|
| RANDOM SAMPLING | 0.112±0.049 | 0.046±0.030 | 0.050±0.002 | 0.036 | 0.048±0.024 | 0.061±0.034 | 0.044±0.019 | 0.078±0.031 | 0.053±0.024 |
| RF+IS | 0.108±0.094 | 0.112±0.034 | 0.086±0.038 | 0.075±0.048 | 0.069±0.046 | 0.062±0.041 | 0.073±0.056 | 0.040 | 0.101±0.047 |
| LR+IS | 0.118±0.046 | 0.091±0.072 | 0.070±0.026 | 0.049±0.028 | 0.138±0.072 | 0.084±0.047 | 0.114±0.077 | 0.065±0.031 | 0.030±0.032 |
| RF+LURE | 0.079±0.042 | 0.166±0.151 | 0.120±0.075 | 0.036 | 0.073±0.031 | 0.040±0.048 | 0.122±0.064 | 0.053±0.025 | 0.099±0.137 |
| LR+LURE | 0.076±0.075 | 0.119±0.076 | 0.073±0.050 | 0.050±0.029 | 0.123±0.089 | 0.046±0.051 | 0.026±0.013 | 0.053±0.025 | 0.053±0.025 |
| *Random Selection + BQ* | | | | | | | | | |
| BQ-RPF RAND | 0.027±0.024 | 0.076±0.054 | 0.044±0.014 | 0.032±0.021 | 0.058±0.046 | 0.066±0.029 | 0.033±0.017 | 0.008±0.004 | 0.030±0.014 |
| BQ-TPF RAND | 0.057±0.038 | 0.048±0.046 | 0.039±0.035 | 0.023±0.007 | 0.042±0.038 | 0.070±0.055 | 0.043±0.024 | 0.016±0.001 | 0.042±0.030 |
| BQ-SF RAND | 0.037±0.011 | **0.020±0.012** | 0.023±0.007 | 0.038±0.018 | 0.037±0.030 | 0.044±0.010 | **0.010±0.005** | 0.019±0.003 | 0.010 |
| *Active Selection + BQ* | | | | | | | | | |
| BQ-RPF | 0.049 | 0.169 | 0.052 | **0.005** | 0.027 | 0.034 | 0.052 | **0.006** | 0.056 |
| BQ-RPF ROUNDED | **0.013** | 0.190 | 0.031 | 0.009 | 0.040 | 0.037 | 0.034 | 0.007 | 0.007 |
| BQ-TPF | 0.073 | 0.167 | 0.025 | 0.013 | **0.001** | **0.004** | 0.070 | 0.008 | 0.048 |
| BQ-TPF ROUNDED | 0.211 | 0.324 | 0.324 | 0.036 | 0.077 | 0.121 | 0.159 | 0.040 | 0.063 |
| BQ-SF | 0.027 | 0.068 | **0.013** | 0.023 | 0.014 | 0.024 | 0.024 | 0.007 | **0.006** |
| BQ-SF ROUNDED | 0.018 | 0.021 | 0.038 | 0.007 | 0.008 | 0.014 | 0.035 | 0.007 | 0.009 |

*Table 8.* MAE (↓) at 1% budget for **GPT-5.1**. ± indicates mean ± std over 5 runs; active BQ is deterministic. Best results are **bolded**.

| METHOD | DICES | DIVE | GQA | GSM8K | JIGSAW | MMLU | STRATEGYQA | SVAMP | TOXICCHAT |
|---|---|---|---|---|---|---|---|---|---|
| RANDOM SAMPLING | 0.139±0.055 | 0.092±0.044 | 0.071±0.051 | 0.032 | 0.055±0.046 | 0.061±0.049 | 0.080±0.024 | 0.064±0.035 | 0.051±0.021 |
| RF+IS | 0.149±0.059 | 0.098±0.048 | 0.093±0.046 | 0.079±0.060 | 0.094±0.081 | 0.057±0.071 | 0.088±0.023 | 0.036 | 0.082±0.069 |
| LR+IS | 0.129±0.052 | 0.075±0.055 | 0.071±0.051 | 0.036±0.004 | 0.105±0.053 | 0.115±0.056 | 0.072±0.042 | 0.093±0.083 | 0.026±0.026 |
| RF+LURE | 0.075±0.045 | 0.079±0.058 | 0.085±0.044 | 0.035±0.004 | 0.100±0.033 | 0.055±0.046 | 0.071±0.031 | 0.050±0.029 | 0.041±0.025 |
| LR+LURE | 0.074±0.052 | 0.036±0.026 | 0.080±0.037 | 0.062±0.061 | 0.129±0.072 | 0.082±0.025 | 0.060±0.037 | 0.050±0.029 | 0.051±0.021 |
| *Random Selection + BQ* | | | | | | | | | |
| BQ-RPF RAND | 0.022±0.017 | 0.114±0.095 | 0.045±0.038 | 0.019±0.012 | 0.050±0.046 | 0.030±0.019 | 0.033±0.014 | 0.034±0.034 | 0.032±0.017 |
| BQ-TPF RAND | 0.053±0.033 | 0.080±0.035 | 0.050±0.066 | 0.055±0.058 | 0.080±0.050 | 0.058±0.044 | 0.065±0.033 | 0.011±0.002 | 0.060±0.017 |
| BQ-SF RAND | 0.018±0.007 | 0.149±0.030 | 0.053±0.015 | 0.041±0.025 | 0.037±0.033 | 0.077±0.027 | **0.010±0.009** | 0.019±0.006 | **0.001** |
| *Active Selection + BQ* | | | | | | | | | |
| BQ-RPF | 0.010 | **0.019** | 0.202 | 0.030 | 0.031 | 0.013 | 0.051 | 0.011 | 0.057 |
| BQ-RPF ROUNDED | **0.001** | 0.057 | 0.105 | **0.004** | 0.061 | 0.025 | 0.016 | **0.003** | 0.023 |
| BQ-TPF | 0.069 | 0.049 | 0.016 | 0.009 | 0.015 | 0.056 | 0.045 | 0.003 | 0.038 |
| BQ-TPF ROUNDED | 0.295 | 0.243 | 0.275 | 0.032 | 0.064 | 0.132 | 0.120 | 0.036 | 0.054 |
| BQ-SF | 0.014 | 0.145 | 0.035 | 0.027 | 0.011 | 0.026 | 0.014 | 0.011 | 0.002 |
| BQ-SF ROUNDED | 0.058 | 0.059 | **0.008** | **0.004** | **0.001** | **0.010** | 0.040 | **0.003** | 0.005 |

*Table 9.* MAE (↓) at 1% budget for **GPT-5.2**. ± indicates mean ± std over 5 runs; active BQ is deterministic. Best results are **bolded**.

| METHOD | DICES | DIVE | GQA | GSM8K | JIGSAW | MMLU | STRATEGYQA | SVAMP | TOXICCHAT |
|---|---|---|---|---|---|---|---|---|---|
| RANDOM SAMPLING | 0.109±0.085 | 0.127±0.068 | 0.071±0.036 | 0.030±0.027 | 0.090±0.031 | 0.138±0.073 | 0.078±0.038 | 0.075±0.020 | 0.049±0.022 |
| RF+IS | 0.056±0.078 | 0.124±0.063 | 0.076±0.037 | 0.100±0.054 | 0.069±0.055 | 0.067±0.030 | 0.058±0.048 | 0.051 | 0.051±0.055 |
| LR+IS | 0.056±0.025 | 0.182±0.065 | 0.093±0.069 | 0.044±0.030 | 0.119±0.089 | 0.091±0.065 | 0.084±0.042 | 0.096±0.071 | 0.049±0.022 |
| RF+LURE | 0.085±0.038 | 0.108±0.072 | 0.044±0.046 | 0.041±0.027 | 0.109±0.061 | 0.053±0.055 | 0.047±0.026 | 0.104±0.068 | 0.035±0.010 |
| LR+LURE | 0.152±0.045 | 0.127±0.068 | 0.062±0.048 | 0.052±0.022 | 0.135±0.065 | 0.110±0.046 | 0.106±0.065 | 0.067±0.020 | 0.040±0.009 |
| *Random Selection + BQ* | | | | | | | | | |
| BQ-RPF RAND | 0.069±0.041 | 0.059±0.056 | 0.054±0.036 | 0.044±0.043 | 0.037±0.025 | 0.121±0.071 | 0.034±0.025 | 0.024±0.029 | 0.032±0.022 |
| BQ-TPF RAND | 0.068±0.074 | 0.059±0.025 | 0.059±0.032 | 0.035±0.022 | 0.093±0.066 | 0.126±0.040 | 0.075±0.045 | 0.027±0.001 | 0.030±0.003 |
| BQ-SF RAND | 0.117 | 0.030±0.015 | 0.042±0.014 | **0.020±0.028** | 0.046±0.029 | 0.130±0.057 | **0.013±0.007** | 0.010±0.005 | 0.008±0.001 |
| *Active Selection + BQ* | | | | | | | | | |
| BQ-RPF | 0.117 | 0.029 | **0.000** | 0.034 | 0.113 | 0.114 | 0.037 | 0.006 | 0.069 |
| BQ-RPF ROUNDED | 0.117 | 0.037 | 0.044 | 0.035 | 0.047 | 0.085 | 0.062 | 0.019 | 0.015 |
| BQ-TPF | **0.014** | 0.076 | 0.007 | 0.032 | **0.024** | 0.039 | 0.180 | 0.021 | 0.027 |
| BQ-TPF ROUNDED | 0.211 | 0.367 | 0.183 | 0.063 | 0.066 | 0.149 | 0.198 | 0.051 | 0.044 |
| BQ-SF | 0.117 | **0.026** | 0.030 | 0.033 | 0.062 | **0.002** | 0.034 | **0.006** | **0.004** |
| BQ-SF ROUNDED | 0.117 | 0.041 | 0.039 | 0.034 | 0.079 | 0.042 | 0.019 | 0.019 | 0.009 |

*Table 10.* MAE (↓) at 1% budget for **Claude 3.5 Haiku**. ± indicates mean ± std over 5 runs; active BQ is deterministic. Best results are **bolded**.

| METHOD | DICES | DIVE | GQA | GSM8K | JIGSAW | MMLU | STRATEGYQA | SVAMP | TOXICCHAT |
|---|---|---|---|---|---|---|---|---|---|
| RANDOM SAMPLING | 0.184±0.076 | 0.106±0.032 | 0.091±0.071 | 0.035±0.001 | 0.059±0.030 | 0.122±0.037 | 0.059±0.036 | 0.125±0.090 | 0.034±0.005 |
| RF+IS | 0.128±0.111 | 0.079±0.054 | 0.091±0.084 | 0.051±0.031 | 0.088±0.087 | 0.084±0.028 | 0.041±0.024 | 0.051 | 0.066±0.041 |
| LR+IS | 0.115±0.140 | 0.081±0.027 | 0.116±0.054 | 0.035±0.001 | 0.128±0.036 | 0.090±0.071 | 0.071±0.044 | 0.067±0.020 | 0.032±0.005 |
| RF+LURE | 0.111±0.090 | 0.148±0.088 | 0.090±0.062 | 0.036±0.001 | 0.130±0.077 | 0.094±0.053 | 0.067±0.033 | 0.067±0.020 | 0.051±0.026 |
| LR+LURE | 0.083±0.067 | 0.068±0.057 | **0.032±0.027** | 0.051±0.027 | 0.105±0.042 | 0.097±0.069 | 0.094±0.061 | 0.067±0.020 | 0.046±0.025 |
| *Random Selection + BQ* | | | | | | | | | |
| BQ-RPF RAND | 0.091±0.093 | 0.093±0.040 | 0.076±0.037 | 0.036±0.021 | 0.060±0.040 | 0.077±0.084 | 0.042±0.025 | 0.031±0.036 | 0.019±0.015 |
| BQ-TPF RAND | 0.071±0.057 | 0.066±0.061 | 0.051±0.041 | 0.039±0.030 | 0.056±0.029 | 0.055±0.043 | 0.044±0.039 | 0.026±0.001 | 0.022±0.006 |
| BQ-SF RAND | 0.073±0.012 | 0.074±0.041 | 0.088±0.022 | 0.027±0.008 | 0.031±0.011 | 0.039±0.019 | 0.009±0.006 | 0.006±0.003 | 0.016±0.001 |
| *Active Selection + BQ* | | | | | | | | | |
| BQ-RPF | **0.011** | 0.242 | 0.036 | **0.007** | 0.050 | 0.068 | 0.039 | 0.006 | 0.085 |
| BQ-RPF ROUNDED | 0.045 | 0.271 | 0.096 | 0.010 | 0.035 | **0.005** | 0.026 | 0.019 | 0.017 |
| BQ-TPF | 0.112 | **0.025** | 0.153 | 0.015 | 0.051 | 0.018 | 0.089 | 0.020 | 0.021 |
| BQ-TPF ROUNDED | 0.100 | 0.332 | 0.432 | 0.037 | 0.004 | 0.231 | 0.175 | 0.051 | 0.038 |
| BQ-SF | 0.078 | 0.051 | 0.055 | 0.022 | **0.001** | 0.036 | **0.002** | **0.006** | 0.012 |
| BQ-SF ROUNDED | 0.027 | 0.067 | 0.069 | 0.009 | 0.035 | 0.037 | 0.018 | 0.019 | **0.001** |

*Table 11.* MAE (↓) at 1% budget for **Claude 3.7 Sonnet**. ± indicates mean ± std over 5 runs; active BQ is deterministic. Best results are **bolded**.

| METHOD | DICES | DIVE | GQA | GSM8K | JIGSAW | MMLU | STRATEGYQA | SVAMP | TOXICCHAT |
|---|---|---|---|---|---|---|---|---|---|
| RANDOM SAMPLING | 0.134±0.060 | 0.087±0.049 | 0.055±0.039 | 0.029 | 0.071±0.032 | 0.101±0.061 | 0.060±0.032 | 0.064±0.038 | 0.050±0.021 |
| RF+IS | 0.167±0.086 | 0.092±0.090 | 0.110±0.070 | 0.052±0.037 | 0.114±0.047 | 0.081±0.040 | 0.071±0.066 | 0.033 | 0.065±0.043 |
| LR+IS | 0.106±0.080 | 0.111±0.107 | 0.053±0.025 | 0.034±0.007 | 0.058±0.036 | 0.120±0.087 | 0.064±0.029 | 0.064±0.038 | 0.023±0.014 |
| RF+LURE | 0.086±0.073 | 0.072±0.061 | 0.031±0.021 | 0.061±0.039 | 0.081±0.038 | 0.079±0.071 | 0.062±0.084 | 0.048±0.031 | 0.077±0.044 |
| LR+LURE | 0.080±0.048 | 0.114±0.078 | 0.056±0.039 | 0.049±0.033 | 0.076±0.051 | 0.070±0.039 | 0.083±0.032 | 0.048±0.031 | 0.043±0.025 |
| *Random Selection + BQ* | | | | | | | | | |
| BQ-RPF RAND | 0.069±0.036 | 0.044±0.028 | 0.072±0.048 | 0.036±0.039 | 0.079±0.032 | 0.034±0.026 | 0.041±0.039 | 0.008±0.006 | 0.015±0.014 |
| BQ-TPF RAND | 0.050±0.070 | 0.081±0.056 | 0.056±0.050 | 0.042±0.032 | 0.039±0.019 | 0.073±0.058 | 0.038±0.030 | 0.008±0.002 | 0.043±0.026 |
| BQ-SF RAND | 0.062±0.032 | 0.048±0.051 | 0.024±0.009 | 0.034±0.006 | 0.049±0.043 | 0.026±0.021 | 0.014±0.008 | 0.025±0.003 | **0.002** |
| *Active Selection + BQ* | | | | | | | | | |
| BQ-RPF | **0.009** | 0.045 | 0.040 | 0.002 | 0.094 | 0.052 | 0.040 | 0.014 | 0.014 |
| BQ-RPF ROUNDED | 0.058 | 0.098 | **0.005** | **0.001** | 0.095 | 0.058 | 0.012 | **0.000** | 0.005 |
| BQ-TPF | 0.038 | **0.002** | 0.054 | 0.007 | 0.061 | 0.090 | 0.006 | 0.001 | 0.035 |
| BQ-TPF ROUNDED | 0.335 | 0.363 | 0.198 | 0.029 | 0.013 | 0.165 | 0.099 | 0.033 | 0.051 |
| BQ-SF | 0.030 | 0.003 | 0.019 | 0.031 | **0.011** | **0.010** | 0.018 | 0.014 | 0.003 |
| BQ-SF ROUNDED | 0.087 | 0.035 | 0.045 | **0.001** | 0.029 | 0.046 | **0.003** | **0.000** | 0.003 |

*Table 12.* MAE (↓) at 1% budget for **Claude 4.5 Sonnet**. ± indicates mean ± std over 5 runs; active BQ is deterministic. Best results are **bolded**.

| METHOD | DICES | DIVE | GQA | GSM8K | JIGSAW | MMLU | STRATEGYQA | SVAMP | TOXICCHAT |
|---|---|---|---|---|---|---|---|---|---|
| RANDOM SAMPLING | 0.132±0.060 | 0.055±0.070 | 0.022±0.014 | 0.027 | 0.062±0.033 | 0.064±0.065 | 0.059±0.036 | 0.063±0.042 | 0.057±0.013 |
| RF+IS | 0.117±0.080 | 0.140±0.086 | 0.131±0.040 | 0.087±0.053 | 0.054±0.035 | 0.059±0.012 | 0.108±0.067 | 0.029 | 0.092±0.080 |
| LR+IS | 0.070±0.043 | 0.081±0.032 | 0.069±0.035 | 0.034±0.008 | 0.027±0.009 | 0.042±0.048 | 0.060±0.038 | 0.063±0.042 | 0.029±0.027 |
| RF+LURE | 0.099±0.077 | 0.062±0.051 | 0.060±0.040 | 0.034±0.008 | 0.064±0.035 | 0.112±0.125 | 0.056±0.028 | 0.063±0.042 | 0.061±0.023 |
| LR+LURE | 0.055±0.074 | 0.126±0.107 | 0.116±0.068 | 0.048±0.034 | 0.084±0.047 | 0.053±0.024 | 0.024±0.028 | 0.046±0.034 | 0.064±0.046 |
| *Random Selection + BQ* | | | | | | | | | |
| BQ-RPF RAND | 0.079±0.042 | 0.130±0.086 | 0.038±0.019 | 0.016±0.012 | 0.047±0.026 | 0.040±0.044 | 0.019±0.022 | 0.015±0.009 | 0.017±0.013 |
| BQ-TPF RAND | 0.123±0.076 | 0.061±0.068 | 0.087±0.057 | 0.023±0.015 | 0.060±0.056 | 0.049±0.027 | 0.059±0.043 | 0.004±0.002 | 0.051±0.025 |
| BQ-SF RAND | 0.181±0.032 | 0.063±0.016 | 0.055±0.014 | 0.041±0.011 | 0.091±0.020 | 0.063±0.021 | 0.037±0.013 | 0.030±0.004 | 0.003 |
| *Active Selection + BQ* | | | | | | | | | |
| BQ-RPF | 0.072 | 0.102 | 0.117 | 0.004 | 0.102 | 0.085 | 0.016 | 0.019 | **0.002** |
| BQ-RPF ROUNDED | 0.049 | 0.140 | 0.064 | **0.000** | 0.122 | 0.050 | **0.004** | 0.004 | 0.009 |
| BQ-TPF | **0.039** | 0.038 | 0.191 | 0.005 | **0.026** | 0.109 | 0.048 | **0.002** | 0.035 |
| BQ-TPF ROUNDED | 0.125 | 0.324 | 0.411 | 0.027 | 0.111 | 0.130 | 0.120 | 0.029 | 0.051 |
| BQ-SF | 0.181 | **0.017** | 0.023 | 0.032 | 0.071 | 0.029 | 0.023 | 0.019 | 0.006 |
| BQ-SF ROUNDED | 0.127 | 0.043 | **0.004** | 0.002 | 0.052 | **0.008** | 0.012 | 0.004 | 0.007 |

*Table 13.* MAE (↓) at 1% budget for **Claude 4.5 Opus**. ± indicates mean ± std over 5 runs; active BQ is deterministic. Best results are **bolded**.

| METHOD | DICES | DIVE | GQA | GSM8K | JIGSAW | MMLU | STRATEGYQA | SVAMP | TOXICCHAT |
|---|---|---|---|---|---|---|---|---|---|
| RANDOM SAMPLING | 0.151±0.052 | 0.049±0.027 | **0.047±0.045** | 0.035 | 0.068±0.038 | 0.039±0.024 | 0.075±0.036 | 0.064±0.036 | 0.060±0.009 |
| RF+IS | 0.160±0.088 | 0.086±0.052 | 0.093±0.054 | 0.055±0.028 | 0.129±0.052 | 0.131±0.056 | 0.112±0.100 | 0.034 | 0.046±0.022 |
| LR+IS | 0.087±0.049 | 0.076±0.052 | 0.098±0.031 | 0.036±0.001 | 0.134±0.081 | 0.114±0.077 | 0.078±0.048 | 0.064±0.036 | 0.037±0.032 |
| RF+LURE | 0.100±0.058 | 0.112±0.074 | 0.135±0.071 | 0.036±0.001 | 0.101±0.042 | 0.109±0.069 | 0.041±0.017 | 0.049±0.030 | 0.038±0.022 |
| LR+LURE | 0.087±0.077 | 0.101±0.061 | 0.096±0.062 | 0.050±0.029 | 0.128±0.063 | 0.067±0.036 | 0.038±0.032 | 0.049±0.030 | 0.056±0.050 |
| *Random Selection + BQ* | | | | | | | | | |
| BQ-RPF RAND | 0.068±0.033 | 0.154±0.082 | 0.083±0.017 | 0.049±0.052 | 0.031±0.023 | 0.024±0.019 | 0.032±0.009 | 0.008±0.005 | 0.032±0.023 |
| BQ-TPF RAND | 0.099±0.066 | 0.070±0.044 | 0.055±0.023 | 0.027±0.012 | 0.105±0.044 | 0.055±0.024 | 0.037±0.022 | 0.010±0.002 | 0.041±0.027 |
| BQ-SF RAND | 0.056±0.011 | 0.041±0.024 | 0.094±0.018 | 0.037±0.014 | 0.056±0.018 | 0.028±0.028 | 0.009±0.006 | 0.021±0.006 | **0.003±0.002** |
| *Active Selection + BQ* | | | | | | | | | |
| BQ-RPF | **0.026** | 0.083 | 0.095 | 0.031 | 0.011 | 0.047 | 0.011 | 0.012 | 0.011 |
| BQ-RPF ROUNDED | 0.073 | 0.115 | 0.126 | **0.006** | 0.081 | 0.061 | 0.005 | **0.001** | 0.004 |
| BQ-TPF | 0.203 | 0.063 | 0.243 | 0.013 | 0.095 | 0.015 | 0.120 | 0.002 | 0.040 |
| BQ-TPF ROUNDED | 0.205 | 0.306 | 0.566 | 0.035 | 0.007 | 0.164 | 0.137 | 0.034 | 0.056 |
| BQ-SF | 0.041 | 0.049 | 0.120 | 0.024 | **0.006** | **0.013** | 0.040 | 0.013 | 0.003 |
| BQ-SF ROUNDED | 0.044 | **0.002** | 0.146 | **0.006** | 0.019 | 0.042 | **0.001** | **0.001** | 0.007 |

*Table 14.* MAE (↓) at 1% budget for **Gemini 2.5 Pro**. ± indicates mean ± std over 5 runs; active BQ is deterministic. Best results are **bolded**.

| METHOD | DICES | DIVE | GQA | GSM8K | JIGSAW | MMLU | STRATEGYQA | SVAMP | TOXICCHAT |
|---|---|---|---|---|---|---|---|---|---|
| RANDOM SAMPLING | 0.060±0.049 | 0.100±0.060 | 0.107±0.072 | 0.049±0.032 | 0.029±0.031 | 0.084±0.060 | 0.043±0.029 | 0.062±0.046 | 0.059±0.049 |
| RF+IS | 0.085±0.069 | 0.087±0.033 | 0.054±0.045 | 0.075±0.048 | 0.117±0.060 | 0.058±0.050 | 0.070±0.053 | 0.024 | 0.066±0.029 |
| LR+IS | 0.083±0.058 | 0.112±0.050 | 0.064±0.056 | **0.039±0.035** | 0.044±0.032 | 0.088±0.010 | 0.040±0.021 | 0.062±0.046 | 0.023±0.027 |
| RF+LURE | 0.072±0.016 | 0.128±0.079 | 0.072±0.052 | 0.045±0.032 | 0.063±0.045 | 0.071±0.027 | 0.044±0.048 | 0.024 | 0.049±0.022 |
| LR+LURE | 0.057±0.047 | 0.060±0.053 | 0.057±0.042 | 0.054±0.035 | 0.055±0.024 | 0.073±0.014 | 0.101±0.073 | 0.043±0.038 | 0.065±0.044 |
| *Random Selection + BQ* | | | | | | | | | |
| BQ-RPF RAND | **0.051±0.021** | 0.135±0.088 | 0.026±0.025 | 0.058±0.038 | 0.063±0.039 | 0.058±0.034 | 0.036±0.032 | 0.020±0.007 | 0.017±0.008 |
| BQ-TPF RAND | 0.123±0.051 | 0.034±0.030 | 0.039±0.026 | 0.059±0.047 | 0.038±0.015 | 0.054±0.045 | 0.079±0.052 | **0.001±0.001** | 0.046±0.027 |
| BQ-SF RAND | 0.172±0.025 | 0.058±0.026 | 0.038±0.018 | 0.054±0.038 | 0.078±0.010 | 0.039±0.018 | 0.025±0.007 | 0.032±0.005 | 0.005 |
| *Active Selection + BQ* | | | | | | | | | |
| BQ-RPF | 0.096 | 0.040 | 0.058 | 0.052 | 0.139 | 0.096 | 0.027 | 0.023 | 0.027 |
| BQ-RPF ROUNDED | 0.134 | 0.069 | 0.024 | 0.086 | 0.137 | 0.057 | 0.010 | 0.009 | **0.001** |
| BQ-TPF | 0.059 | **0.005** | 0.113 | 0.059 | 0.096 | 0.054 | **0.000** | 0.008 | 0.042 |
| BQ-TPF ROUNDED | 0.145 | 0.299 | 0.152 | 0.154 | **0.025** | 0.143 | 0.084 | 0.024 | 0.057 |
| BQ-SF | 0.184 | 0.052 | 0.025 | 0.057 | 0.051 | 0.036 | 0.004 | 0.023 | 0.002 |
| BQ-SF ROUNDED | 0.121 | 0.034 | **0.002** | 0.124 | 0.030 | **0.022** | 0.011 | 0.009 | 0.006 |

*Table 15.* MAE (↓) at 1% budget for **Gemini 3 Flash**. ± indicates mean ± std over 5 runs; active BQ is deterministic. Best results are **bolded**.

| METHOD | DICES | DIVE | GQA | GSM8K | JIGSAW | MMLU | STRATEGYQA | SVAMP | TOXICCHAT |
|---|---|---|---|---|---|---|---|---|---|
| RANDOM SAMPLING | 0.133±0.060 | 0.168±0.084 | 0.075±0.056 | 0.033±0.004 | 0.095±0.024 | 0.058±0.047 | 0.060±0.033 | 0.064±0.038 | 0.041±0.028 |
| RF+IS | 0.129±0.084 | 0.085±0.074 | 0.123±0.058 | 0.083±0.062 | 0.051±0.045 | 0.036±0.039 | 0.088±0.070 | 0.033 | 0.042±0.022 |
| LR+IS | 0.066±0.041 | 0.152±0.050 | 0.034±0.018 | 0.038±0.004 | 0.055±0.071 | 0.058±0.039 | 0.049±0.043 | 0.064±0.038 | 0.021±0.027 |
| RF+LURE | 0.080±0.035 | 0.133±0.087 | 0.056±0.038 | 0.045±0.018 | 0.064±0.047 | 0.080±0.028 | 0.046±0.012 | 0.048±0.031 | 0.053±0.028 |
| LR+LURE | 0.053±0.045 | 0.074±0.056 | 0.071±0.063 | 0.033±0.004 | 0.095±0.048 | 0.048±0.036 | 0.057±0.007 | 0.048±0.031 | 0.055±0.050 |
| *Random Selection + BQ* | | | | | | | | | |
| BQ-RPF RAND | 0.038±0.041 | 0.070±0.047 | 0.062±0.038 | 0.028±0.011 | 0.048±0.041 | 0.053±0.044 | 0.038±0.023 | 0.008±0.006 | 0.027±0.016 |
| BQ-TPF RAND | 0.068±0.044 | 0.127±0.037 | 0.074±0.049 | 0.035±0.018 | 0.077±0.030 | 0.035±0.027 | 0.052±0.031 | 0.008±0.002 | 0.045±0.030 |
| BQ-SF RAND | 0.053±0.032 | 0.066±0.043 | 0.022±0.016 | 0.026±0.003 | 0.060±0.015 | 0.070±0.011 | 0.045±0.006 | 0.025±0.003 | 0.007 |
| *Active Selection + BQ* | | | | | | | | | |
| BQ-RPF | 0.050 | 0.239 | 0.042 | **0.000** | 0.164 | 0.066 | 0.056 | 0.014 | 0.012 |
| BQ-RPF ROUNDED | **0.000** | 0.227 | 0.021 | 0.005 | 0.149 | 0.026 | 0.033 | **0.000** | 0.004 |
| BQ-TPF | 0.169 | 0.132 | 0.100 | 0.009 | 0.069 | 0.092 | 0.033 | 0.001 | 0.043 |
| BQ-TPF ROUNDED | 0.070 | 0.158 | 0.422 | 0.031 | 0.055 | 0.112 | 0.110 | 0.033 | 0.059 |
| BQ-SF | 0.008 | 0.090 | 0.024 | 0.028 | 0.035 | 0.049 | 0.035 | 0.014 | **0.002** |
| BQ-SF ROUNDED | 0.038 | **0.005** | **0.002** | 0.003 | **0.014** | **0.010** | **0.020** | **0.000** | 0.003 |

*Table 16.* MAE (↓) at 1% budget for **Gemini 3 Pro**. ± indicates mean ± std over 5 runs; active BQ is deterministic. Best results are **bolded**.

| METHOD | DICES | DIVE | GQA | GSM8K | JIGSAW | MMLU | STRATEGYQA | SVAMP | TOXICCHAT |
|---|---|---|---|---|---|---|---|---|---|
| RANDOM SAMPLING | 0.121±0.049 | 0.059±0.035 | 0.128±0.099 | 0.063±0.057 | 0.094±0.032 | 0.104±0.064 | 0.120±0.032 | 0.198±0.206 | 0.050 |
| RF+IS | 0.117±0.071 | 0.111±0.042 | 0.108±0.073 | 0.092±0.030 | 0.068±0.047 | 0.094±0.056 | 0.123±0.073 | 0.149±0.085 | 0.067±0.072 |
| LR+IS | 0.093±0.068 | 0.126±0.089 | 0.082±0.066 | 0.063±0.035 | 0.121±0.114 | 0.103±0.052 | 0.034±0.030 | 0.170±0.105 | 0.050±0.030 |
| RF+LURE | 0.072±0.016 | 0.215±0.182 | 0.070±0.026 | 0.078±0.058 | 0.153±0.080 | 0.110±0.112 | 0.075±0.043 | 0.153±0.087 | 0.037±0.016 |
| LR+LURE | 0.081±0.076 | 0.086±0.049 | 0.070±0.050 | 0.096±0.051 | 0.109±0.067 | 0.081±0.024 | 0.045±0.035 | 0.111±0.051 | 0.017 |
| *Random Selection + BQ* | | | | | | | | | |
| BQ-RPF RAND | 0.059±0.034 | 0.057±0.037 | 0.069±0.052 | 0.149±0.071 | 0.048±0.046 | **0.043±0.027** | 0.059±0.059 | 0.118±0.083 | 0.032±0.018 |
| BQ-TPF RAND | 0.120±0.027 | 0.096±0.040 | 0.093±0.066 | 0.088±0.067 | 0.090±0.074 | 0.105±0.048 | 0.075±0.027 | 0.094±0.062 | 0.049±0.026 |
| BQ-SF RAND | 0.070±0.014 | **0.019±0.011** | 0.023±0.009 | 0.323±0.022 | 0.135±0.039 | 0.104±0.048 | 0.116±0.013 | 0.236±0.019 | **0.001±0.001** |
| *Active Selection + BQ* | | | | | | | | | |
| BQ-RPF | 0.031 | 0.117 | 0.090 | **0.014** | 0.070 | 0.109 | 0.112 | 0.114 | 0.027 |
| BQ-RPF ROUNDED | 0.067 | 0.157 | 0.099 | 0.149 | 0.139 | 0.149 | **0.015** | 0.234 | 0.008 |
| BQ-TPF | **0.022** | 0.205 | 0.089 | 0.014 | 0.087 | 0.266 | 0.164 | **0.047** | 0.034 |
| BQ-TPF ROUNDED | 0.203 | 0.517 | 0.395 | 0.181 | **0.002** | 0.478 | 0.304 | 0.296 | 0.050 |
| BQ-SF | 0.073 | 0.020 | **0.020** | 0.298 | 0.100 | 0.120 | 0.114 | 0.180 | 0.003 |
| BQ-SF ROUNDED | 0.107 | 0.039 | 0.039 | 0.343 | 0.109 | 0.195 | 0.140 | 0.127 | 0.011 |

*Table 17.* MAE (↓) at 1% budget for **Gemma 3 12B**. ± indicates mean ± std over 5 runs; active BQ is deterministic. Best results are **bolded**.

| METHOD | DICES | DIVE | GQA | GSM8K | JIGSAW | MMLU | STRATEGYQA | SVAMP | TOXICCHAT |
|---|---|---|---|---|---|---|---|---|---|
| RANDOM SAMPLING | 0.056±0.031 | 0.105±0.032 | 0.056±0.026 | 0.054±0.023 | 0.051±0.029 | 0.154±0.114 | 0.097±0.058 | 0.095±0.075 | 0.050±0.021 |
| RF+IS | 0.063±0.027 | 0.140±0.029 | 0.065±0.051 | 0.110±0.061 | 0.082±0.040 | 0.129±0.077 | 0.080±0.034 | 0.046 | 0.048±0.034 |
| LR+IS | 0.061±0.032 | 0.118±0.082 | 0.086±0.074 | 0.054±0.023 | 0.218±0.083 | 0.058±0.033 | 0.050±0.058 | 0.066±0.025 | 0.037±0.018 |
| RF+LURE | 0.059±0.034 | 0.090±0.065 | 0.115±0.073 | 0.040±0.020 | 0.111±0.091 | 0.072±0.039 | 0.070±0.037 | 0.127±0.138 | 0.037±0.018 |
| LR+LURE | 0.059±0.034 | 0.111±0.104 | 0.069±0.020 | 0.060±0.026 | 0.138±0.085 | 0.126±0.082 | 0.114±0.088 | 0.066±0.025 | 0.052 |
| *Random Selection + BQ* | | | | | | | | | |
| BQ-RPF RAND | 0.064±0.074 | 0.075±0.064 | 0.059±0.046 | 0.044±0.027 | 0.083±0.037 | 0.083±0.065 | 0.046±0.040 | 0.011±0.007 | 0.026±0.017 |
| BQ-TPF RAND | **0.025±0.015** | 0.049±0.016 | 0.089±0.087 | 0.032±0.017 | 0.103±0.062 | 0.104±0.090 | 0.083±0.058 | 0.020±0.002 | 0.045±0.026 |
| BQ-SF RAND | 0.117 | 0.020±0.010 | 0.076±0.025 | 0.006±0.005 | 0.125±0.022 | 0.094±0.051 | 0.018±0.011 | 0.012±0.004 | 0.001±0.001 |
| *Active Selection + BQ* | | | | | | | | | |
| BQ-RPF | 0.070 | 0.020 | 0.095 | 0.030 | **0.043** | 0.055 | 0.024 | **0.000** | 0.014 |
| BQ-RPF ROUNDED | 0.108 | 0.030 | 0.011 | 0.029 | 0.142 | **0.037** | 0.034 | 0.013 | **0.000** |
| BQ-TPF | 0.133 | 0.241 | **0.003** | 0.034 | 0.139 | 0.054 | 0.016 | 0.142 | 0.034 |
| BQ-TPF ROUNDED | 0.094 | 0.325 | 0.370 | 0.056 | 0.443 | 0.431 | 0.148 | 0.046 | 0.052 |
| BQ-SF | 0.117 | 0.023 | 0.051 | **0.002** | 0.142 | 0.097 | **0.010** | 0.000 | 0.007 |
| BQ-SF ROUNDED | 0.117 | **0.012** | 0.059 | 0.028 | 0.162 | 0.150 | 0.026 | 0.013 | 0.018 |

*Table 18.* MAE (↓) at 1% budget for **Gemma 3 27B**. ± indicates mean ± std over 5 runs; active BQ is deterministic. Best results are **bolded**.

| METHOD | DICES | DIVE | GQA | GSM8K | JIGSAW | MMLU | STRATEGYQA | SVAMP | TOXICCHAT |
|---|---|---|---|---|---|---|---|---|---|
| RANDOM SAMPLING | 0.063±0.044 | — | — | 0.073±0.061 | 0.079±0.026 | 0.119±0.050 | 0.061±0.072 | 0.130±0.109 | 0.029±0.019 |
| RF+IS | 0.067±0.043 | — | — | 0.132±0.150 | 0.075±0.051 | 0.092±0.063 | 0.076±0.069 | 0.070±0.008 | 0.111±0.071 |
| LR+IS | 0.069±0.040 | — | — | 0.166±0.089 | 0.042±0.032 | 0.081±0.028 | 0.049±0.043 | 0.102±0.052 | 0.027±0.027 |
| RF+LURE | 0.082±0.066 | — | — | 0.135±0.093 | 0.073±0.057 | 0.069±0.063 | 0.088±0.056 | 0.098±0.054 | 0.055±0.051 |
| LR+LURE | 0.090±0.072 | — | — | 0.100±0.011 | 0.017±0.027 | 0.056±0.051 | 0.103±0.069 | 0.070±0.008 | 0.042±0.024 |
| *Random Selection + BQ* | | | | | | | | | |
| BQ-RPF RAND | 0.111±0.067 | — | — | 0.106±0.046 | 0.077±0.053 | 0.056±0.040 | 0.058±0.036 | 0.053±0.036 | 0.010±0.004 |
| BQ-TPF RAND | 0.084±0.047 | — | — | 0.142±0.056 | 0.059±0.040 | 0.094±0.052 | 0.061±0.057 | 0.058±0.023 | 0.042±0.018 |
| BQ-SF RAND | 0.041 | — | — | 0.141±0.076 | 0.045±0.037 | 0.070±0.038 | 0.032±0.016 | 0.021±0.005 | **0.001±0.001** |
| *Active Selection + BQ* | | | | | | | | | |
| BQ-RPF | **0.031** | — | — | **0.009** | 0.016 | 0.068 | 0.069 | 0.034 | 0.018 |
| BQ-RPF ROUNDED | 0.037 | — | — | 0.122 | 0.030 | 0.107 | 0.093 | 0.047 | 0.003 |
| BQ-TPF | 0.177 | — | — | 0.084 | 0.006 | 0.319 | 0.079 | 0.048 | 0.036 |
| BQ-TPF ROUNDED | 0.115 | — | — | 0.143 | 0.138 | 0.406 | 0.073 | 0.080 | 0.053 |
| BQ-SF | 0.041 | — | — | 0.012 | **0.003** | **0.019** | **0.022** | **0.012** | 0.003 |
| BQ-SF ROUNDED | 0.041 | — | — | 0.072 | 0.038 | 0.038 | 0.057 | 0.046 | 0.013 |

*Table 19.* MAE (↓) at 1% budget for **Qwen 3 32B**. ± indicates mean ± std over 5 runs; active BQ is deterministic. Best results are **bolded**. The model doesn't support image inputs so we don't have DIVE and GQA numbers.

C.3.2. ABLATION STUDY ON DIFFERENT EMBEDDING MODELS

The embedding of prompt features influence the quality of PROEVAL. We conducted an ablation study on BQ-RPF with different embedding models (Google `gemini_embedding_001`, OpenAI `text_embedding_3_small`, OpenAI `text_embedding_3_large`, and `all_minilm_l6_v2`), using a sampling budget of 50.

We evaluated Gemini 2.5 Flash on StrategyQA using BQ-RPF, Section C.3.2 shows a clear trend: models with larger embedding dimensions tend to yield lower MAEs. That means, using stronger embedding models tend to have a more accurate performance estimation.

| Embedding Model | Dimension | Tier | BQ MAE |
|---|---|---|---|
| `gemini_embedding_001` | 3072 | Strong | 0.0425 |
| `text_embedding_3_large` | 3072 | Strong | 0.0654 |
| `text_embedding_3_small` | 1536 | Medium | 0.0834 |
| `all_minilm_l6_v2` | 384 | Weak | 0.0900 |

*Table 20.* A preliminary experiment shows that PROEVAL using embedding models with larger embedding dimensions tend to yield lower MAEs.

## C.4. Additional Results on Failure Case Discovery

We further evaluate generalization across target models in Table 21. Our methods consistently outperform Rand-Gen across all five targets, with the advantage being most pronounced on stronger models that are harder to break: on GPT 5, TSS-TPF discovers 18.9% failures on StrategyQA versus 5.1% for Rand-Gen, a 3.7× improvement. As expected, weaker target models yield higher overall failure rates (e.g., Gemma3 at 67.8% vs. GPT 5 at 18.9% for TSS-TPF on StrategyQA), but the relative benefit of guided generation remains substantial across all difficulty levels.

| Target Model | StrategyQA | | | | GSM8K | | | |
|---|---|---|---|---|---|---|---|---|
| | TSS-TPF | TSS-RPF | SS-Gen-RPF | Rand-Gen | TSS-TPF | TSS-RPF | SS-Gen-RPF | Rand-Gen |
| *Gemma3 (27b)* | **67.8%** | 65.4% | 60.2% | 20.5% | 57.4% | 55.1% | **70.3%** | 15.4% |
| *Qwen3 (32b)* | 44.2% | 42.1% | **45.3%** | 15.2% | 42.5% | 40.2% | **50.6%** | 12.1% |
| *Gemini 2.5 Flash* | 33.0% | 26.0% | **40.3%** | 30.3% | 20.7% | **31.3%** | 27.7% | 7.3% |
| *Claude 3.7 Sonnet* | **24.6%** | 22.5% | 20.1% | 8.3% | 20.4% | 18.7% | **25.2%** | 5.4% |
| *GPT 5* | **18.9%** | 17.2% | 15.4% | 5.1% | 16.8% | 15.3% | **20.5%** | 3.2% |

*Table 21*. Failure discovery rates for different target models using TSS-TPF, TSS-RPF, SS-Gen-RPF, and Rand-Gen on generating StrategyQA and GSM8K problems. We evaluate 5 different target models with Gemini 3 Pro as the generator, using 100 iterations. TSS-TPF and SS-Gen-RPF consistently outperform the Rand-Gen baseline.

**A note on query generator validity.** The reported failure rates could be influenced by mistakes made by the query generator when synthesizing "harder" problems. Hence we chose a strong model, Gemini 3 Pro, as the generator, and it is aligned with the widely used LLM-as-a-judge setup. All methods in our synthesis experiments (Rand-Gen, SS-Gen, TSS) use the same query generator under identical temperature settings. Therefore, the relative performance delta remains a valid indicator of PROEVAL's ability to surface model-specific vulnerabilities.

| Generator Model | StrategyQA | | | | GSM8K | | | |
|---|---|---|---|---|---|---|---|---|
| | TSS-TPF | TSS-RPF | SS-Gen-RPF | Rand-Gen | TSS-TPF | TSS-RPF | SS-Gen-RPF | Rand-Gen |
| *Gemini 3 Pro* | 33.0% | 26.0% | **40.3%** | 30.3% | 20.7% | **31.3%** | 27.7% | 7.3% |
| *GPT 5* | **43.2%** | 41.5% | 39.8% | 15.1% | 28.3% | 26.1% | **35.2%** | 8.4% |
| *GPT 4o* | **38.7%** | 36.1% | 35.2% | 12.4% | 24.5% | 22.8% | **31.6%** | 5.8% |
| *Gemini 3 Flash* | 33.5% | **35.8%** | 31.2% | 10.7% | 19.3% | 18.1% | **27.4%** | 4.5% |
| *Qwen3 (32b)* | **30.4%** | 28.7% | 27.1% | 8.3% | 16.2% | 14.8% | **22.5%** | 3.4% |
| *Gemma3 (27b)* | 24.6% | **26.3%** | 23.8% | 6.5% | 12.7% | 11.4% | **18.9%** | 2.6% |

*Table 22*. Comparison of failure discovery rates for TSS-TPF, TSS-RPF, SS-Gen-RPF, and Rand-Gen across six query generator models using Gemini 2.5 Flash as the target model. Results are reported as the percentage of failures discovered, evaluated over 100 iterations.

To quantify the accuracy of the query generator's answers for its generated queries, we conducted a small human verification study on 80 randomly sampled questions from the failure discovery task (generated by Rand-Gen, SS-Gen and TSS), covering grade school math (GSM8K-style) and implicit reasoning (StrategyQA-style)[3].

We found that Gemini 3 Pro answered 90% of these questions correctly. For the 8 questions it answered incorrectly, we inspected the answers given by the target model Gemini 2.5 Flash. On 5 of those questions, the target model gave the same incorrect answer as Gemini 3 Pro; on 2 questions, it gave a different but still incorrect answer; and on 1 question (where the answer should be yes or no), it gave the correct answer but used flawed reasoning. Overall, this human study demonstrates that our estimated failure rate is a lower bound of the true failure rate. Intuitively, this makes sense because a "weaker" model will very likely fail on problems that a "stronger" model failed at.

The LLM generator's quality impacts the effectiveness of the methods SS-Gen and TSS. For example, if an LLM keeps generating the same token, methods like SS-Gen and TSS will certainly not work. So it is important to choose a capable LLM as the generator. As shown in Table 22, stronger generators consistently yield higher failure discovery rates. For instance, TSS-RPF discovers 41.5% failures on StrategyQA with GPT 5, compared to 28.7% with Qwen3 (32b) and 26.3% with Gemma3 (27b). Note that our approach reduces the number of queries necessary for the LLM generator to discover a

---

[3]The four co-authors solved the questions independently to validate the answers generated by the models.

new failure. So even under a limited budget, one may select a more expensive model as LLM generator to be used with our method.

**Generation evaluation problems.** To provide a qualitative understanding of the differences between the discovery strategies, Tables 23 and 24 present side-by-side (SxS) examples of questions generated by the baseline *Random Generation* and our *Active Generation* (TSS) strategies. While random generation often produces simpler, direct questions that target models easily solve (Score 0.0), the active strategy consistently discovers complex failure cases (Score 1.0).

| Random Generation | Active Generation |
|---|---|
| *Score: 0.0 (Solved)* 
 Is dopamine snorted nasally by drug users? | *Score: 1.0 (Failure)* 
 Would the animal representing the first year of the 21st century be able to physically complete the race mentioned in the Great Race of the Zodiac myth? |
| *Score: 0.0 (Solved)* 
 Could the Austrian casualties from Seven Years' War fit in Indianapolis Motor Speedway? | *Score: 1.0 (Failure)* 
 Can a person use a mechanical typewriter to write a document while the room is completely dark? |
| *Score: 0.0 (Solved)* 
 Was a person sold a Creative Commons License for Boticelli's The Birth of Venus ripped off? | *Score: 1.0 (Failure)* 
 Can a person complete a physical jigsaw puzzle in total darkness if they have already memorized the image? |
| *Score: 0.0 (Solved)* 
 Do Republicans reject all forms of welfare? | *Score: 1.0 (Failure)* 
 Can a person perform work on a physical object if the object is being moved solely by an electric motor powered by a fuel cell? |
| *Score: 0.0 (Solved)* 
 Would a giant panda find the primary ingredient of a traditional Neapolitan pizza suitable for its diet? | *Score: 1.0 (Failure)* 
 If a gourmet chef serves a dish containing 'bee bread' to a strict vegetarian, has the chef violated the diner's dietary restrictions? |
| *Score: 0.0 (Solved)* 
 Would an expensive tailor use adhesive to create a shorter hem on slacks? | *Score: 1.0 (Failure)* 
 Can a diesel-powered generator produce electricity if it is completely out of fuel? |
| *Score: 0.0 (Solved)* 
 Do Youtube viewers get unsolicited audiobook advice often? | *Score: 1.0 (Failure)* 
 Would a proofreader for the world's most popular video game magazine in 1993 have needed to review coverage of the Commodore 64? |
| *Score: 0.0 (Solved)* 
 Did England win any Olympic gold medals in 1800? | *Score: 1.0 (Failure)* 
 Could a proofreader for the world's most popular magazine in 1964 have reviewed an article about the premiere of the film 'The Sound of Music'? |
| *Score: 0.0 (Solved)* 
 Did J. D. Salinger ever ask his father for a quinceañera? | *Score: 1.0 (Failure)* 
 Did the author of the poem 'The Children's Hour' also write a collection of stories about a fictional village in the Acadian region of Canada? |

*Table 23.* Example StrategyQA-like reasoning problems generated by Random vs. Active strategies. Active sampling strategy consistently synthesizes "multi-hop" constraints that require bridging disparate knowledge domains.

| Random Generation | Active Generation |
|---|---|
| *Score: 0.0 (Solved)* | *Score: 1.0 (Failure)* |
| A whirligig spins at five times the speed of a thingamabob. A whatchamacallit spins eleven times faster than a thingamabob. A whatchamacallit spins at 121 meters per second. How fast does a whirligig spin? | A metal rod is heated at a constant rate of 15 degrees per minute. However, the rod loses heat to the environment at a rate proportional to its current temperature relative to the room: for every 10 degrees it is above the starting temperature of 70 degrees, it loses an additional 2 degrees of heating efficiency per minute. If the rod is heated for exactly 4 minutes, but the cooling loss only triggers at the start of each new minute based on the temperature at the end of the previous minute, what is the final temperature of the rod? |
| *Score: 0.0 (Solved)* | *Score: 1.0 (Failure)* |
| Zoey and Sydney are having a watermelon seed spitting contest. Whoever spits their seeds the most total distance wins. They each get one watermelon. Zoey's has 40 seeds and she spits each one 10 feet. Sydney's has 35 she spits each one 12 feet. What is the average total distance spat? | A custom rug designer is creating a large rectangular rug that is 12 feet wide and 18 feet long. The rug features a solid border around the entire edge that is 2 feet wide. Inside that border, the remaining space is divided into four identical square panels. If the rest of the interior space not covered by these four squares is filled with decorative fringe, how many square feet of decorative fringe are there? |
| *Score: 0.0 (Solved)* | *Score: 1.0 (Failure)* |
| Sarah has a rope that is 20 meters long. Her friend wants to buy the rope for $2 a meter. Sarah plans to use the profit to buy a new rope, which at the store costs $1.5 a meter. How much money will she have left over after she buys the new rope? | In five years, Peter will be twice as old as Quinton was three years ago. If Quinton is currently half as old as Peter was when Quinton was born, and Peter is 40 years old now, how old is Quinton? |
| *Score: 0.0 (Solved)* | *Score: 1.0 (Failure)* |
| Farmer Ben has 5 apple trees. Each tree produces 40 apples. He decides to keep 25 apples for his family and sell the rest at the local market. If he sells the apples in bags of 5 and charges $3 per bag, how much money will he make? | A trivia contest has 20 questions. For every correct answer, a player earns 10 points. For every incorrect answer, they lose 5 points. For every question left unanswered, they lose 2 points. Sarah answered 15 questions in total. If she had 3 times as many correct answers as incorrect answers, but the score multiplier of 1.5x only applies to the total points earned from correct answers before deductions, what is her final score? |
| *Score: 0.0 (Solved)* | *Score: 1.0 (Failure)* |
| Farmer Leo has a rectangular garden that is 12 feet long and 8 feet wide. He wants to put a wooden fence around the entire perimeter, but he needs to leave a 4-foot gap for a gate. If the fencing material costs $5 per foot, how much will it cost Leo to buy the fencing he needs? | Oliver is making custom jelly bean jars for a party. He has a large bag containing 300 jelly beans. He fills 8 small jars with 25 jelly beans each. He then eats 15 jelly beans while cleaning up. If he wants to split the remaining jelly beans equally between 2 large bowls, how many jelly beans will be in each bowl? |

*Table 24.* Example GSM8K-like math problems generated by Random vs. Active strategies. Active sampling strategy consistently discovers harder problems requiring multi-step reasoning, system of equations, or layered constraints.

# D. Generation Prompts for Failure Discovery

This appendix details the prompts used by our test case generators. All methods share a common base structure, with differences in whether they include: (1) topic constraints, (2) anchor examples from previously identified hard problems and (3) using tuned prompt features.

| Method | is Generation | Topic | Anchor Examples | Tuned Prompt Features |
|---|---|---|---|---|
| Rand | ✗ | ✗ | ✗ | ✗ |
| SS-RPF | ✗ | ✗ | ✓ | ✗ |
| SS-TPF | ✗ | ✗ | ✓ | ✓ |
| Rand-Gen | ✓ | ✗ | ✗ | ✗ |
| Rand-T-Gen | ✓ | ✓ | ✗ | ✗ |
| Rand-Anchor-Gen | ✓ | ✗ | ✓ | ✗ |
| SS-Gen-RPF | ✓ | ✗ | ✓ | ✗ |
| SS-Gen-TPF | ✓ | ✗ | ✓ | ✓ |
| TSS-RPF | ✓ | ✓ | ✓ | ✗ |
| TSS-TPF | ✓ | ✓ | ✓ | ✓ |

*Table 25.* Comparison of failure discovery methods across different components.

## D.1. GSM8K Generator Prompts

### D.1.1. RAND-GEN

Rand-Gen (No Topic, No Anchors)

```
Generate a creative grade-school math word problem (GSM8K-style).

Requirements:
- Problem must require 2-3 steps of arithmetic to solve
- Answer must be a specific number
- Make the problem clear and unambiguous

IMPORTANT: You MUST solve the problem step-by-step yourself
BEFORE providing the answer. Show your work in the "solution"
field to verify the ground_truth is correct.

Output JSON format: {"question": "...", "solution": "Step 1: ...
Step 2: ... Therefore the answer is X", "ground_truth": <number>}
```

### D.1.2. RAND-T-GEN

Rand-T-Gen (With Topic, No Anchors)

```
Generate a creative grade-school math word problem (GSM8K-style).

TOPIC TO USE: {selected_topic_label}

Requirements:
- Problem must require 2-3 steps of arithmetic to solve
- Problem should be related to the topic above
- Answer must be a specific number
- Make the problem clear and unambiguous

IMPORTANT: You MUST solve the problem step-by-step yourself
BEFORE providing the answer. Show your work in the "solution"
field to verify the ground_truth is correct.

Output JSON format: {"question": "...", "solution": "Step 1: ...
Step 2: ... Therefore the answer is X", "ground_truth": <number>}
```

### D.1.3. RAND-ANCHOR-GEN

Rand-Anchor-Gen (Random Anchors, No Topic)

```
Generate a creative grade-school math word problem (GSM8K-style)
that is similar to the following example problems.

=== HARD EXAMPLE PROBLEMS (AI models failed on these) ===
{anchor_context}

Requirements:
- Problem must require 2-3 steps of arithmetic to solve
- Answer must be a specific number
- Make the problem clear and unambiguous

IMPORTANT: You MUST solve the problem step-by-step yourself
BEFORE providing the answer. Show your work in the "solution"
field to verify the ground_truth is correct.

Output JSON format: {"question": "...", "solution": "Step 1: ...
Step 2: ... Therefore the answer is X", "ground_truth": <number>}
```

## D.1.4. SS-GEN

SS-Gen (No Topic, With Anchors)

```
Generate a creative grade-school math word problem (GSM8K-style).

=== HARD EXAMPLE PROBLEMS (AI models failed on these) ===
{anchor_context}

Requirements:
- Problem must require 2-3 steps of arithmetic to solve
- MIMIC THE REASONING PATTERN of the hard examples above
- Answer must be a specific number
- Make the problem clear and unambiguous

IMPORTANT: You MUST solve the problem step-by-step yourself
BEFORE providing the answer. Show your work in the "solution"
field to verify the ground_truth is correct.

Output JSON format: {"question": "...", "solution": "Step 1: ...
Step 2: ... Therefore the answer is X", "ground_truth": <number>}
```

## D.1.5. TSS

TSS (With Topic, With Anchors)

```
Generate a creative grade-school math word problem (GSM8K-style).

TOPIC TO USE: {selected_topic_label}

=== HARD EXAMPLE PROBLEMS (AI models failed on these) ===
{anchor_context}

Requirements:
- Problem must require 2-3 steps of arithmetic to solve
- Problem should be related to the topic above
- MIMIC THE REASONING PATTERN of the hard examples above
- Answer must be a specific number
- Make the problem clear and unambiguous

IMPORTANT: You MUST solve the problem step-by-step yourself
BEFORE providing the answer. Show your work in the "solution"
field to verify the ground_truth is correct.

Output JSON format: {"question": "...", "solution": "Step 1: ...
Step 2: ... Therefore the answer is X", "ground_truth": <number>}
```

**D.2. StrategyQA Generator Prompts**

D.2.1. RAND-GEN

Rand-Gen (No Topic, No Anchors)

```
Generate a creative yes/no (StrategyQA-style) reasoning question.

Requirements:
- Question must require 2-3 steps of multi-hop reasoning
- Answer must be YES or NO only
- Make the question clear and unambiguous

IMPORTANT: You MUST reason through the answer step-by-step
yourself BEFORE providing it. Show your reasoning in the
"reasoning" field to verify the ground_truth is correct.

Output JSON format: {"question": "...", "reasoning": "Step 1: ...
Step 2: ... Therefore the answer is YES/NO",
"ground_truth": "yes" or "no"}
```

D.2.2. RAND-T-GEN

Rand-T-Gen (With Topic, No Anchors)

```
Generate a creative yes/no (StrategyQA-style) reasoning question.

TOPIC TO USE: {selected_topic_label}

Requirements:
- Question must require 2-3 steps of multi-hop reasoning
- Question should be related to the topic above
- Answer must be YES or NO only
- Make the question clear and unambiguous

IMPORTANT: You MUST reason through the answer step-by-step
yourself BEFORE providing it. Show your reasoning in the
"reasoning" field to verify the ground_truth is correct.

Output JSON format: {"question": "...", "reasoning": "Step 1: ...
Step 2: ... Therefore the answer is YES/NO",
"ground_truth": "yes" or "no"}
```

### D.2.3. RAND-ANCHOR-GEN

```
Rand-Anchor-Gen (Random Anchors, No Topic)

Generate a creative yes/no (StrategyQA-style) reasoning question.
that is similar to the following example problems.

=== HARD EXAMPLE PROBLEMS (AI models failed on these) ===
{anchor_context}

Requirements:
- Question must require 2-3 steps of multi-hop reasoning
- MIMIC THE REASONING PATTERN of the hard examples above
- Answer must be YES or NO only
- Make the question clear and unambiguous

IMPORTANT: You MUST reason through the answer step-by-step
yourself BEFORE providing it. Show your reasoning in the
"reasoning" field to verify the ground_truth is correct.

Output JSON format: {"question": "...", "reasoning": "Step 1: ...
Step 2: ... Therefore the answer is YES/NO",
"ground_truth": "yes" or "no"}
```

### D.2.4. SS-GEN

```
SS-Gen (No Topic, With Anchors)

Generate a creative yes/no (StrategyQA-style) reasoning question.

=== HARD EXAMPLE QUESTIONS (AI models failed on these) ===
{anchor_context}

Requirements:
- Question must require 2-3 steps of multi-hop reasoning
- MIMIC THE REASONING PATTERN of the hard examples above
- Answer must be YES or NO only
- Make the question clear and unambiguous

IMPORTANT: You MUST reason through the answer step-by-step
yourself BEFORE providing it. Show your reasoning in the
"reasoning" field to verify the ground_truth is correct.

Output JSON format: {"question": "...", "reasoning": "Step 1: ...
Step 2: ... Therefore the answer is YES/NO",
"ground_truth": "yes" or "no"}
```

D.2.5. TSS-GEN

TSS-Gen (With Topic, With Anchors)

```
Generate a creative yes/no (StrategyQA-style) reasoning question.

TOPIC TO USE: {selected_topic_label}

=== HARD EXAMPLE QUESTIONS (AI models failed on these) ===
{anchor_context}

Requirements:
- Question must require 2-3 steps of multi-hop reasoning
- Question should be related to the topic above
- MIMIC THE REASONING PATTERN of the hard examples above
- Answer must be YES or NO only
- Make the question clear and unambiguous

IMPORTANT: You MUST reason through the answer step-by-step
yourself BEFORE providing it. Show your reasoning in the
"reasoning" field to verify the ground_truth is correct.

Output JSON format: {"question": "...", "reasoning": "Step 1: ...
Step 2: ... Therefore the answer is YES/NO",
"ground_truth": "yes" or "no"}
```

## D.3. Variable Descriptions

- {anchor_context}: A formatted list of 3-5 example problems that the target model previously answered incorrectly, selected via our Superlevel Set acquisition strategy.

- {selected_topic_label}: A topic label extracted via BERTopic clustering (e.g., "shopping, prices, discounts" for GSM8K or "history, war, presidents" for StrategyQA).

# E. Source Data Selection Methods for Bayesian Quadrature

## E.1. Problem Setting

Under Assumption 1 and 2, we have access to historical datasets $\mathcal{D} = \{D_i\}_{i=1}^{N}$ where each $D_i = \{(x_j, y_{ij})\}_{j=1}^{M_i}$ contains evaluation scores $y_{ij} \sim \mathcal{N}(f_i(x_j), \sigma^2)$ from model $i$ on question $x_j$. The key insight is that $\boldsymbol{y}_i = [y_{ij}]_{j=1}^{M}$ forms a sample from a multivariate Gaussian $\mathcal{N}(\boldsymbol{u}, \Sigma)$.

**The Source Selection Problem:** Given a new target model $f_*$ and a limited evaluation budget, which subset of source models $\mathcal{S} \subseteq \{1, \dots, N\}$ should we use to estimate the prior $(\hat{\boldsymbol{u}}, \hat{\Sigma})$?

**Why This Matters:**

- Using all models assumes all $\boldsymbol{y}_i$ come from the same GP prior, which is violated when the target is out-of-distribution (OOD).

- Using irrelevant models corrupts the prior estimate, leading to poor BQ predictions.

- Optimal selection balances having enough data against including dissimilar models.

## E.2. Source Selection Methods

### E.2.1. LEAVE-ONE-OUT PRIOR (BASELINE)

**Goal:** Use all available historical models *excluding the target* to estimate the GP prior, without further selection.

**Mathematical Formulation:** Given $N$ historical models with the target as model $*$, the source set includes all except the target:

$$\mathcal{S} = \{1, \dots, N\} \setminus \{*\} \tag{13}$$

The prior estimates become:

$$\hat{\boldsymbol{u}} = \frac{1}{|\mathcal{S}|} \sum_{i \in \mathcal{S}} \boldsymbol{y}_i, \quad \hat{\Sigma} = \frac{1}{|\mathcal{S}| - 1} \sum_{i \in \mathcal{S}} (\boldsymbol{y}_i - \hat{\boldsymbol{u}})(\boldsymbol{y}_i - \hat{\boldsymbol{u}})^\top \tag{14}$$

> **Important: Target Always Excluded**
>
> The target model is never included in the source set. This prevents data leakage and ensures the prior is estimated from independent data only.

**Intuition:** This is the simplest approach—no selection, just use everything except the target. It works well when all models share similar behavior, but may include outliers that corrupt the prior.

**Abstention Rule:** Can combine with Spearman correlation-based abstention:

$$\rho_{\max} = \max_{i \in \mathcal{S}} \text{spearman}(\boldsymbol{y}_*, \boldsymbol{y}_i) \tag{15}$$

where $\boldsymbol{y}_*$ and $\boldsymbol{y}_i$ are per-question prediction vectors. If $\rho_{\max} < \tau$, abstain.

### E.2.2. GMM CLUSTERING

**Goal:** Identify models with similar "behavior profiles" by clustering in feature space.

**Mathematical Formulation:** Let $\Phi = [\phi_1, \dots, \phi_N]^\top \in \mathbb{R}^{N \times d}$ be the feature matrix where $\phi_i$ represents model $i$'s behavior (e.g., PCA-reduced per-question predictions on reference benchmarks). Fit a Gaussian Mixture Model with $K$ components:

$$p(\phi) = \sum_{k=1}^{K} \pi_k \cdot \mathcal{N}(\phi \mid \mu_k, \Sigma_k) \tag{16}$$

Select $K$ via BIC minimization:

$$\text{BIC}(K) = -2 \log p(\Phi \mid \theta_K) + d_K \log N \tag{17}$$

where $d_K$ is the number of free parameters. The source selection becomes:

$$\mathcal{S} = \{i : z_i = z_*\} \setminus \{*\} \tag{18}$$

where $z_i = \arg\max_k p(z_i = k \mid \phi_i)$ is the cluster assignment.

**Intuition:** Models in the same cluster have similar prediction patterns on reference benchmarks, suggesting they sample from a common GP prior.

**Abstention Rule:** Abstain if the GMM selects fewer than `min_sources` models (default=3). In 78 experiments, there were 17 abstentions (22%) when the cluster had $< 3$ models. This dramatically improves reliability—mean MAE drops from $0.0394 \to 0.0274$.

### E.2.3. CORRELATION-BASED SELECTION

**Goal:** Select models whose per-question predictions are highly correlated with the target.

**Mathematical Formulation:** Use Spearman rank correlation for robustness to outliers: $\rho_{i*} = \text{spearman}(\boldsymbol{y}_i, \boldsymbol{y}_*)$. Select top-$k$ models by Spearman correlation with the target:

$$\mathcal{S} = \text{top-}k\{i : \rho_{i*}\} \tag{19}$$

### E.2.4. MAHALANOBIS DISTANCE SELECTION

**Goal:** Select models geometrically closest to the target in the feature distribution.

**Mathematical Formulation:** Given feature vectors $\phi_i$ and empirical covariance $\hat{\Sigma}_\Phi$:

$$d_{\text{Mah}}(\phi_i, \phi_*) = \sqrt{(\phi_i - \phi_*)^\top \hat{\Sigma}_\Phi^{-1}(\phi_i - \phi_*)} \tag{20}$$

Select top-$k$ models with the smallest distance.

**OOD Detection Variant:** Test if target is OOD via $d^2_{\text{Mah}}(\phi_*, \hat{\mu}_\Phi) \sim \chi^2_d \implies$ abstain if

$$d^2_{\text{Mah}}(\phi_*, \hat{\mu}_\Phi) > \chi^2_{d,1-\alpha}.$$

### E.2.5. LEAVE-ONE-OUT LIKELIHOOD SELECTION

**Goal:** Select models that are "typical" under the overall model distribution.

**Mathematical Formulation:** For each candidate model $i$, fit a Gaussian on all other models:

$$\mu_{-i} = \frac{1}{N-1}\sum_{j\neq i}\phi_j, \quad \Sigma_{-i} = \frac{1}{N-2}\sum_{j\neq i}(\phi_j - \mu_{-i})(\phi_j - \mu_{-i})^\top \tag{21}$$

Compute log-likelihood $\ell_i = \log\mathcal{N}(\phi_i \mid \mu_{-i}, \Sigma_{-i})$ and select the top-$k$ models with the highest $\ell_i$.

### E.2.6. HYPOTHESIS TEST SELECTION

**Goal:** Select models that statistically cannot be distinguished from the target.

**Mathematical Formulation:** For target $\phi_*$ and candidate $\phi_i$, test if they come from the same distribution using Hotelling's $T^2$:

$$T^2 = (\phi_* - \phi_i)^\top \hat{\Sigma}_{\text{combined}}^{-1}(\phi_* - \phi_i) \sim \chi^2_d \tag{22}$$

Source Selection: $\mathcal{S} = \{i : p\text{-value}(T_i^2) > \alpha\}$. *Note:* In our experiments, Hotelling's $T^2$ had no power with $d = 12$ features and $n = 2$ samples, producing identical results across candidates.

### E.2.7. MARDIA'S MULTIVARIATE NORMALITY TEST

**Goal:** Select models such that combined data satisfies the GP assumption of joint normality.

**Mathematical Formulation:** For combined data $X = [\phi_*, \phi_i]^\top$, compute Mardia's skewness and kurtosis. Select top-$k$ models by combined $p$-value (Fisher's method):

$$\chi^2_{\text{combined}} = -2(\log p_{\text{skew}} + \log p_{\text{kurt}}) \sim \chi^2_4 \tag{23}$$

*Note:* The combined normality assumption rarely holds in practice—this method proved too strict for practical abstention.

### E.3. Experimental Results

**Experimental Setup:** We evaluated across 6 benchmarks (`gsm8k`, `svamp`, `strategyqa`, `mmlu`, `jigsaw`, `toxicchat`) using 13-15 LLMs per benchmark. Features were PCA-reduced per-question predictions retaining 95% variance ($\sim$12 components). The metric is the Mean Absolute Error (MAE) between the BQ-SF estimate and true accuracy at 20 evaluation iterations.

*Table 26.* Overall Performance (78 Experiments, MAE at Iteration 20). Sorted by median MAE (lower is better).

| Method | N | Abstain | Mean MAE | Median MAE | Std MAE |
|---|---|---|---|---|---|
| GMM + min≥3 | 61 | 17 | 0.0274 | **0.0109** | 0.0421 |
| GMM (no abstention) | 78 | 0 | 0.0394 | 0.0125 | 0.0584 |
| LOO Prior + spearman≥0.7 | 36 | 42 | 0.0170 | 0.0120 | 0.0175 |
| LOO Prior | 78 | 0 | 0.0277 | 0.0131 | 0.0386 |
| LOO Prior + spearman≥0.5 | 52 | 26 | 0.0235 | 0.0134 | 0.0301 |
| Mardia | 78 | 0 | 0.0406 | 0.0145 | 0.0610 |
| Hypothesis Test | 78 | 0 | 0.0284 | 0.0147 | 0.0363 |
| Mahalanobis | 78 | 0 | 0.0526 | 0.0152 | 0.1071 |
| LOO Likelihood | 78 | 0 | 0.0660 | 0.0158 | 0.1315 |
| Correlation | 78 | 0 | 0.0482 | 0.0162 | 0.0973 |
| Random | 78 | 0 | 0.0408 | 0.0316 | 0.0355 |

*Table 27.* Pearson vs Spearman Correlation Comparison.

| Correlation | $\tau$ | N | Abstain % | Mean MAE | Median MAE | Std MAE |
|---|---|---|---|---|---|---|
| Pearson | 0.3 | 70 | 10% | 0.0269 | 0.0131 | 0.0375 |
| Spearman | 0.3 | 68 | 13% | 0.0281 | 0.0131 | 0.0383 |
| Pearson | 0.5 | 57 | 27% | 0.0287 | 0.0152 | 0.0371 |
| Spearman | 0.5 | 52 | 33% | 0.0235 | 0.0134 | 0.0301 |
| Pearson | 0.7 | 42 | 46% | 0.0239 | 0.0134 | 0.0321 |
| Spearman ⋆ | 0.7 | 36 | 54% | 0.0170 | 0.0120 | 0.0175 |
| Pearson | 0.9 | 27 | 65% | 0.0137 | 0.0085 | 0.0141 |
| Spearman | 0.9 | 22 | 72% | 0.0131 | 0.0099 | 0.0112 |

**Key Observations:**

1. **GMM with abstention achieves best overall performance:** GMM + min≥3 yields the best median MAE (0.0109). It abstains when the cluster has fewer than 3 models (22% of cases).

2. **Abstention dramatically improves GMM reliability:** Key insight is that low-selection cases (1-2 models) are failure modes. Abstention drops the mean MAE by 30% and Std by 28%.

3. **Spearman-based abstention for LOO Prior achieves lowest variance:** Spearman is more robust to outliers in per-question predictions than Pearson. At $\tau \geq 0.7$, it achieves the lowest standard deviation (0.0175), though with a 54% abstention rate.

4. **Advanced selection methods underperform:** Methods selecting fewer models (top-$k = 5$) have higher variance than using all models. Over-selectivity means that when the selection is wrong, the limited sources provide insufficient data to form a stable prior.

### E.4. Recommendations

*Table 28.* Method Comparison Summary.

| Method | Median MAE | Std | Abstain | Recommendation |
|---|---|---|---|---|
| GMM + min$\geq$3 | **0.0109** | 0.0421 | 22% | Best overall choice |
| GMM (no abstention) | 0.0125 | 0.0584 | 0% | High risk/reward |
| LOO Prior | 0.0131 | 0.0386 | 0% | Safe default |
| LOO Prior + corr$\geq$0.7 | 0.0134 | **0.0321** | 46% | Lowest variance |
| Hypothesis Test | 0.0147 | 0.0363 | 0% | Best among advanced |
| Random | 0.0316 | 0.0355 | 0% | Baseline only |

Based on the verified results, **GMM + min$\geq$3** abstention is the optimal choice for combining good selection with explicit failure detection. If abstention is not acceptable for the pipeline, the **LOO Prior** baseline serves as a safe default.

```
# Recommended Pipeline Implementation
from auto_data_selection import auto_select_with_abstention

source_models, should_abstain = auto_select_with_abstention(
    reference_benchmarks, target_model, data_dir, min_sources=3
)

if should_abstain:
    # Fall back to a default estimate or skip prediction
    pass
else:
    # Run BQ with selected source models
    pass
```

**Future Directions:** Future improvements could include learning adaptive abstention thresholds per benchmark category, ensembling multiple selection methods, online adaptation as evaluations arrive, and deriving theoretical regret bounds for these selection strategies.

