# OpenReview forum: "ProEval: Proactive Failure Discovery and Efficient Performance Estimation for Generative AI Evaluation"
_ICML.cc/2026/Conference — ICML 2026 regular_

### Official Review · Reviewer_763h · 2026-02-24

**Soundness:** 3
**Presentation:** 3
**Significance:** 3
**Originality:** 3
**Overall Recommendation:** 4
**Confidence:** 3

**Summary:**

In the paper, the authors propose ProEval, a proactive method to efficiently evaluate generative models and discover potential failure cases. ProEval uses a gaussian process surrogate to predict the performance patterns of the tested model, and proactively discover failure cases by targeted sampling and synthesis-based augmentation. Experiments show that ProEval has superior performance in failure discovery rates and efficiency.

**Compliance With Llm Reviewing Policy:**

Affirmed.

**Final Justification:**

The authors' rebuttal addressed most of my concerns. I maintain my original score.

**Key Questions For Authors:**

None

**Limitations:**

The paper does not contain an explicit "limitation" section. The authors should discuss potential limitations of the proposed method from perspectives such as:
- Cases that the assumptions involved in the method fail, and related consequences (e.g., performance drop)
- Cases that the priors fail to generalize, and related consequences (e.g., performance and efficiency drops)
- Modules that may fail in inference (e.g., the LLM generator), and their impacts on method performance

**Strengths And Weaknesses:**

Strengths:
- Interesting and novel viewpoint of solution. The method unifies large-scale model evaluation and failure discovery with the same GP-based predictions.
- High generalizability and flexibility of the method. The embedding-based priors are specifically designed for transfer learning and generalizations to other benchmarks, which increases the flexibility of the method.
- Strong theoretical proof and empirical evaluations. The theoretical proof related to GP priors and the experimental results under multiple experimental settings validate the effectiveness of the method.
- Clear paper structure and writing.

Weaknesses:
- Lack of analyses on assumption 1. The high efficiency of the method is mostly dependent on assumption 1. Though with Figure 2, it is still not clear whether the assumption holds in real-world deployment. There is a lack of discussions about specific cases that the assumption does not hold, performance of the method in such cases, etc.
- Heavy dependence on priors. As stated in Section 3.2, BQ converges after only 1-2 evaluation iterations. This typically means that the tested model is fundamentally similar to other models already tested on the benchmark. This phenomenon varies under different benchmarks and tested models, and the efficiency of the method may reduce severely if the tested models are of much different accuracies, or the current tested model behaves differently.
- Lack of experiments on the LLM generator. There is no experiment about the impacts of the LLM generator to the failure discovery performance, or details of hyper-parameters of the LLM generator (e.g., prompt structure). When the LLM generator fails to generate high-quality samples, the effectiveness of the method may reduce.

---

> ### Author Rebuttal · Authors · 2026-03-30
>
> We thank the reviewer for acknowledging ProEval’s novelty, flexibility, strong theoretical grounding and clear writing. We address each concern below:
>
>
> > W1: Lack of analyses on Assumption 1 in real-world deployment… “There is a lack of discussions about specific cases that the assumption does not hold, performance of the method in such cases, etc.”
>
>
> Our experiments already constitute a realistic setting, where a new model is introduced and needs to be evaluated on established benchmarks. Our setup involves 16 production-grade LLMs evaluated on 9 established benchmarks spanning math reasoning, safety alignment and classification. We’d be happy to engage in more discussions if the reviewer could clarify the kind of real-world deployment they have in mind.
>
>
> Regarding the validity of Assumption 1: In Figure 2, we visualize the sample covariance matrices on StrategyQA, GSM8K, MMLU and SVAMP, computed across 5 models. The visualization confirms that the model's performance scores are both positively and negatively correlated across questions, which illustrates that Assumption 1 can hold in practice.
>
>
> We acknowledge that there exist scenarios where Assumption 1 does not hold, but we respectfully disagree that there is a “lack of discussions about specific cases that the assumption does not hold, performance of the method in such cases” in our submission. In Section 2.2 (last paragraph), we detailed how to select the source data for pre-training so that Assumption 1 is valid. In Section 3.2, we presented that there was not enough source data for G0 and O1 (i.e., Assumption 1 did not hold for them) with a detailed case study in §C.2, and showed their performance in Figure 4. We also deepened the analysis in Figure 7, which shows that including unrelated models leads to higher MAEs for models like G1, G2, and C1. These are concrete experiment results analyzing and deliberately constructing cases where Assumption 1 does not hold.
>
>
>
>
>
>
> > W2: Heavy prior dependence, BQ converges in 1-2 steps .
>
>
> Please see our response to Reviewer h2vp’s W1.
>
>
>
>
> > W3: experiments on the LLM generator, no details of hyper-parameters (e.g., prompt structure)
>
>
> Details of prompt structure: Section D presents the prompt structures for the LLM generator.
>
>
> We experimented with different prompt structures and they correspond to different strategies (SS-Gen and TSS) for failure discovery. The results can be found in Section 3.3.
>
>
> Impact of LLM generators: We acknowledge that the LLM generator’s quality impacts the effectiveness of the methods SS-Gen and TSS. For example, if an LLM keeps generating the same token, methods like SS-Gen and TSS will certainly not work. So it is important to choose a capable LLM as the generator. Note that our approach reduces the number of queries necessary for the LLM generator to discover a new failure. So even under a limited budget, one may select a more expensive model as LLM generator to be used with our method.
>
>
> We conducted new experiments testing four different generators with Gemma 3 (27B) as the target model. By generating 90 problems, we compared the failure discovery rates of Rand-Gen and TSS on GSM8K and StrategyQA tasks. The results below show that in general the stronger model has a better performance in failure discovery.
>
>
>
>
> | Generator Model | StrategyQA (TSS) | StrategyQA (Rand-Gen) | GSM8K (TSS) | GSM8K (Rand-Gen) |
> | :--- | :---: | :---: | :---: | :---: |
> | **gemini3_flash** | 31.1% | 30.0% | 37.8% | 28.9% |
> | **gpt_4o**        | 28.9% | 16.7% | 35.6% | 17.7% |
> | **qwen3_32b**     | 26.7% | 13.3% | 33.3% | 10.0% |
> | **gemma3_27b**    | 23.3% | 14.4% | 20.0% | 6.7% |
>
>
>
>
> > Limitations
>
>
> As mentioned under W1, the submitted paper contains detailed investigations on the validity of the assumption, source data selection approach to ensure the validity, and the consequence of not having enough source data to learn a GP prior. For the LLM generator, we will add the limitation discussion (with contents stated under W3).

---

> > ### Author Rebuttal · Reviewer_763h · 2026-04-03
> >
> > My concerns are fully addressed. I maintain my original score.

---

### Official Review · Reviewer_jauK · 2026-03-11

**Soundness:** 3
**Presentation:** 3
**Significance:** 3
**Originality:** 3
**Overall Recommendation:** 4
**Confidence:** 3

**Summary:**

This paper proposes ProEval, a framework for sample-efficient evaluation of generative AI systems that combines pre-trained Gaussian-process surrogates with two downstream objectives: aggregate performance estimation via Bayesian quadrature and failure discovery via superlevel-set sampling. The method supports two transfer settings, one using historical per-item evaluation statistics on the same benchmark and another using learned embeddings for cross-benchmark transfer, and it further extends failure discovery with topic-aware LLM-based test synthesis.

**Compliance With Llm Reviewing Policy:**

Affirmed.

**Final Justification:**

The authors have adequately addressed my main concerns. The clarification on the normalization typo in Equation (4) is reasonable, and the updated proof for Theorem 2 focusing on the discrete empirical mean resolves the ambiguity around the dependence on p(x). The justification for choosing a Gaussian observation model over GP classification (computational tractability for closed-form BQ posteriors) is pragmatic, and the GPC experiment results help contextualize this design choice. I appreciate the authors' commitment to fixing Figure 10 and citing the suggested related work.

**Key Questions For Authors:**

Refer to Weaknesses

**Limitations:**

yes

**Strengths And Weaknesses:**

Strengths

- Evaluating generative AI models is increasingly expensive, and a framework that reduces evaluation cost by 10–100× addresses a genuine need.

Weaknesses

- The central estimator is mathematically inconsistent across the paper. In Equation (1), S is defined as an expectation under p(x). Equation (4) approximates it as a raw sum without 1/M normalization. Yet in Appendix B, the proof suddenly introduces the 1/M factor, contradicting the main text. The reported MAE values (order 1e-2) only make sense with normalization since benchmarks have M (700-2000) with binary scores. Beyond this normalization issue, the proof of Theorem 2 has additional gaps: the dependence on p(x) disappears without justification in the integral bound, and the condition N >> M is not met in practice (N~15, M up to 1600). Since the theory is advertised as a main contribution, these inconsistencies are a serious concern.

- The Gaussian observation model is misspecified for binary failure data. The performance scores are explicitly binary (0/1), yet the GP surrogate assumes Gaussian observation noise. GP classification with a Bernoulli/probit link would be the standard choice for binary outcomes. The BQ Rounded variant implicitly acknowledges this mismatch, but its performance is erratic — dramatically worse than plain BQ on MMLU (0.098 vs. 0.003) while better on DICES (0.003 vs. 0.035).

- Figure 10 has overlapping text (Just Suggestion)

- Some relevant recent work is missing.
[1] Adaptive Prediction-Powered AutoEval with Reliability and Efficiency Guarantees.
[2] Cer-Eval: Certifiable and Cost-Efficient Evaluation Framework for LLMs

---

> ### Author Rebuttal · Authors · 2026-03-30
>
> > W1: (a) The central estimator is mathematically inconsistent across the paper.. (b) the proof of Theorem 2 has additional gaps: the dependence on p(x) disappears without justification in the integral bound..(c) the condition N >> M is not met in practice..
>
>
> (a)
> We thank the reviewer for their meticulous reading and for pointing out the notation inconsistency. We clarify that the $1/M$ normalization is consistently used throughout our implementation, experiments and the proof in Appendix B. The omission of the normalization factors in Equation (4) was a typo error during final formatting and page-limit adjustments. As the reviewer correctly noted, our reported MAE values ($10^{-2}$) reflect this normalized scale. We will correct the equations to include the $1/M$ factor for the estimated expectation of $S$ and $1/M^2$ for the estimated variance of $S$ to ensure full consistency across the manuscript. This is a purely notational correction and does not change any reported experimental results.
>
>
> (b)
> We thank the reviewer for this insightful observation. The initial notation using $p(x)$ was intended to represent a general BQ framework; however, in the context of GenAI evaluation, the practical target is the empirical mean over a discrete evaluation set $\{x_1, \dots, x_M\}$. We acknowledge that some intermediate steps and assumptions were underspecified in the original text.
>
>
> To resolve any ambiguity, we have updated the proof to focus explicitly on the bound between our estimator $\hat{S}_t$ and the ground truth empirical mean $S_t$:
>
>
> $|\hat S_t - S_t| = \left| \frac{1}{M} \sum_{j=1}^M (\hat \mu_t(x_j) - \mu_t(x_j)) \right| \leq \frac{1}{M} \sum_{j=1}^M |\hat\mu_t(x_j) - \mu_t(x_j)|.$
>
>
> The explicit dependence on $p(x)$ is removed so we can directly align the theory with the discrete nature of benchmark scoring. We emphasize that this refinement clarifies the mathematical derivation without changing the core theoretical results or the validity of our bounds.
>
>
>
>
> (c) Our empirical results demonstrate that even when the theoretical bound is loose, the estimation accuracy remains high. We believe our theoretical analyses are critical contributions for the reasons mentioned in our answer to Reviewer cbcc’s W1(a) and that the unbiasedness of our estimator is true regardless of the scale of $N$ or $M$.
>
>
> >  W2: The Gaussian observation model is misspecified for binary failure data.
>
>
> The choice of the observation model is a deliberate trade-off to ensure computational tractability and analytical updates in BQ with pre-trained GPs.
>
>
> While a Bernoulli/probit link is standard for binary data, it precludes a closed-form posterior for the integral $S$ and the pre-training objective. This would require approximate inference (e.g., variational inference, MCMC), which introduces approximation errors that may defeat the purpose of accurately modeling binary data.
>
>
> We experimented with a standard GP classifier (GPC) using Laplace approximation (described in Rasmussen&Williams, “Gaussian Processes for Machine Learning”, 2026) and the approximated log marginal likelihood  for the pre-training. In the settings of Table 1 at a 1% budget, GPC often underperforms, yielding an MAE of 0.1653 on StrategyQA and 0.0516 on SVAMP. We believe more research, especially on transfer learning for GPC, is required to make concrete claims about how to best use GPC for our tasks. Such research, however, exceeds the scope of this conference paper.
>
>
> The "BQ Rounded" variant’s results, while varied, provide a baseline for future work in non-Gaussian BQ for GenAI evaluation.
>
>
>
>
> > W3: Figure 10 has overlapping text
>
>
> Thank you for pointing this out. We will try to reduce the overlapping and write the exact clustered models in text.
>
>
> > W4: relevant recent work
>
>
> Thank you for the recommendations. We will cite these papers.

---

> > ### Author Rebuttal · Reviewer_jauK · 2026-04-02
> >
> > I will keep my positive score.

---

### Official Review · Reviewer_cbcc · 2026-03-12

**Soundness:** 3
**Presentation:** 3
**Significance:** 3
**Originality:** 3
**Overall Recommendation:** 4
**Confidence:** 3

**Summary:**

ProEval is a framework for sample-efficient evaluation of generative models that uses a Gaussian Process prior, constructed via transfer learning from historical evaluation or pre-trained text/image embeddings. The paper avoids the need for large-scale evaluations by using Bayesian quadrature to estimate performance metrics and showcases ProEvals ability to discover diverse failures using superlevel set sampling. The paper uses a large number of LLMs/VLMs and demonstrates ProEval’s success across multiple benchmarks.

**Compliance With Llm Reviewing Policy:**

Affirmed.

**Key Questions For Authors:**

Please see the Strengths and Weaknesses for questions to clarify confusion and issues.

**Limitations:**

yes

**Strengths And Weaknesses:**

Strengths
1. The use of Bayesian Quadature for performance estimation and failure discovery as superlevel set estimation, all within the Gaussian Process framework, is principled and is an improvement over prior work
2. Case I (Transfer via empirical statistics) and Case II (Transfer via learned embeddings) are both highly relevant and are practical scenarios
3. The paper demonstrates ProEval’s efficiency, requiring very few samples compared to alternative methods under both Case I and Case II
4. The thorough testing of the 16 different models, the different types of evaluation, as well as the ablation studies, provides credence to ProEval
5. Section 3.2’s experiment performance estimation shows that including misaligned source models degrades performance, and that the proposed Gaussianity-based filtering prevents this

Weaknesses
1. In Theorem 2, the paper states that ‘assume N ≫ t’ and ‘The bound is non-vacuous if N (the number of historical models) grows at a faster rate than M (the number of test samples).’ However, in the experiments, there are 16 models (LLMs and VLMs), but in Appendix C, Table 5, the sample size is much larger than the number of models. This raises questions about the usefulness of the bounds. Additionally, the number of samples is a small fraction of the standard size, but no details are provided on how the samples were chosen. Furthermore, does ProEval scale with more samples?
2. The paper uses a linear kernel on embeddings and can outperform other methods; however, because it’s linear, it’s limited to capturing linear relationships. If there are nonlinear relationships, then the kernel will be a poor approximation. In Section 2.2, it is stated that the Matérn Kernel can be used for transfer learning and to define embeddings; however, the Matérn Kernel is only used in BQ vanilla. It would be interesting to see whether the method performs even better with the Matérn kernel.
3. All experiments use the text-embedding-3-large embedding model from OpenAI as the fixed embedding in the GP encoder. There is no sensitivity analysis of the embedding quality, nor any study on the impact of different embedding models across the 16 LLMs and VLMs.
4. In Figure 4, BQ Rounded and BQ consistently outperform Random; however, most experiments show that BQ Rounded shows lower MAE than BQ except in G2, O3, O5, C1, and C2. However, in Figures 5 and 6 and in Section 3.3, the paper states that Gemini 2.5 Flash (G2) is chosen. Still, there is no discussion of this choice or any repetition of the experiments with other models to see how ProEval changes with respect to the chosen model. Additionally, BQ Rounded sometimes outperforms BQ in Table 1. Can the authors provide additional context to clarify this?
5. In Section 3.3, the experiment uses the Query Generator to both generate a question and answer it as the ground truth. However, there is no validation of the Query Generator’s results, as the Query Generator can make mistakes and produce incorrect solutions; the target model can solve the question correctly, but ProEval would claim a failure of the target model. While TSS and SS-Gen perform well, evaluating the correctness of the query generator is important for validating their quality. Additionally, the choice of Gemini 3 Pro as the query generator and Gemini 2.5 Flash seems arbitrary.

---

> ### Author Rebuttal · Authors · 2026-03-30
>
> > W1: (a) …the sample size is much larger than the number of models..the usefulness of the bounds.. (b) no details are provided on how the samples were chosen. (c) does ProEval scale with more samples?
>
>
> (a)
> Our theoretical analyses on the bounds are useful for the following reasons: (1) during model development, it is likely that the number of models to evaluate ($N$) is larger than the number of datapoints in a validation or test set, because of the sweep over learning rate, batch size, architecture variants, regularizations, etc. Each config of those parameters corresponds to multiple models since there are many model checkpoints during training. (2) Our bound provides a worst-case guarantee and characterizes how error scales with the scale of the kernel ($\kappa$) and the number of models ($N$), offering the insight that increasing $N$ can lead to better estimation. These insights are not straightforward to obtain without theoretical analyses.
>
>
> The difference between practice and theory is not unique to our paper. Works like “Gaussian Process Optimization in the Bandit Setting: No Regret and Experimental Design” by Srinivas et al., and “Pre-trained Gaussian Processes for Bayesian Optimization” by Wang et al. have also shown that gaps between practical performance and theoretical guarantees exist, but the theoretical results are still valuable contributions.
>
>
> (b)
> Samples were chosen through active selection using the argmax of acquisition functions. The details on how samples are chosen are in Section 2.3 (for performance evaluation) and Section 2.4 (for proactive failure discovery).
>
>
> (c)
> Yes, ProEval scales with more samples. In the submission, Section C.3 presents the additional results for larger scales of samples. More samples generally lead to lower MAE, but ProEval converges very quickly.
>
>
> > W2: Matérn kernel used only in BQ Vanilla
>
>
> In Table 2, we used the Matérn kernel for all BQ-based methods to ensure a fair comparison. We will make this clear in the paper.
>
>
>
>
> > W3: No sensitivity analysis on embedding models
>
>
> Please see our reply to Reviewer h2vp’s W2.
>
>
> > W4: In Figure 4, .. Can the authors provide additional context to clarify this?
>
>
>
>
> If we understand correctly, the reviewer is asking (a) why we chose Gemini 2.5 Flash (G2) as the target model in the experiment, (b) what the performance of ProEval is for other target models and (c) the context of “BQ Rounded sometimes outperforms BQ in Table 1”.
>
>
> (a) We envision a scenario where a new model is introduced, there are not a lot of evaluation results, and efficient evaluation / failure discovery is needed. G2 was a relatively recent model at the time of our experiments, and people may have such evaluation needs for it. Moreover, in our experiments, ProEval’s performance on G2 is quite representative of other models in both performance evaluation and failure discovery tasks. These are the reasons we used G2 as the target model in some figures / tables.
>
>
> (b) The aggregated results of ProEval for performance estimation on other models can be found in Figure 4. We have the individual experiment results used for aggregation, and will include them in the appendix. For failure discovery, we will include results for other models. Below are failure discovery rates using Gemini 3 Pro as the generator model and 90 samples in total for 4 other target models. Our results show that TSS is consistently better than Rand-Gen.
>
>
> | Target Model | StrategyQA (TSS) | StrategyQA (Rand-Gen) | GSM8K (TSS) | GSM8K (Rand-Gen) |
> | :--- | :---: | :---: | :---: | :---: |
> | *gemma3_27b* | 46.7% | 28.9% | 27.8% | 3.3% |
> | *qwen3_32b* | 16.7% | 13.3% | 63.3% | 26.7% |
> | *gpt_4o* | 14.4% | 10.0% | 32.2% | 6.7% |
> | *claude37_sonnet* | 15.6% | 6.7% | 17.8% | 3.3% |
>
>
>  (c) BQ Rounded rounds the posterior estimates to {0, 1} before summation. Even if the posterior mean for an input $x$ is close to 0.5, BQ Rounded will still set the value to be either 0 or 1, which can introduce noise in the estimated performance score. This can make BQ Rounded’s performance not very stable.
>
>
>
>
> > W5: ..the Query Generator can make mistakes and produce incorrect solutions,,
>
>
>
>
> In the widely used LLM-as-judge paradigm, which is similar to our setting, the standard practice is to select the strongest available model. At the time of our experiments, Gemini 3 Pro was state-of-the-art. (See the answer above for why we chose Gemini 2.5 Flash to demonstrate the results.)
>
>
> The focus of this work is to efficiently discover the failure cases, not to generate the right answer or replace human rating. All methods in our comparison use the same generator under identical conditions and so the relative comparison is fair. If human raters are ultimately needed to verify discovered failure, ProEval still reduces the number of samples that need human review compared to random exploration. We will add these clarifications to the paper.

---

> > ### Author Rebuttal · Reviewer_cbcc · 2026-04-03
> >
> > I thank the authors for their response. My remaining concerns revolve around W2 and W5. For W2, the authors claim that in Table 2 the Matérn kernel was used for all BQ cases; however, I believe this to be in conflict with what the paper states. In Section 3.2, BQ is defined as using transfer learning, and the transfer learning construction in Section 2.2 is built entirely on linear kernels. If the kernel choice differs between Table 1 and Table 2, this needs to be explicitly stated and justified, as it changes the interpretation of the results. For W5, I understand the authors' point that all methods use the same generator and therefore the relative comparison is fair. However, I remain concerned that the reported failure rates may be inflated, particularly the TSS, SS-Gen cases, which by design generate harder problems that are also more likely to produce incorrect ground truths from the query generator. Without any verification of ground-truth accuracy the failure discovery results should be interpreted with caution. I suggest the authors to perform a small human audit to verify that the generator's ground truths are valid. I maintain my score.

---

> > > ### Author Response · Authors · 2026-04-06
> > >
> > > Thank you for your careful review and for providing further clarification on your concerns regarding W2 and W5. We appreciate your continued engagement and the opportunity to address these points more precisely.
> > >
> > > > W2: Kernel Consistency in Transfer Learning (Table 2 vs. Section 2.2)
> > >
> > > We clarify that the Matérn kernel was used for all BQ cases in Table 2 (Case II) to ensure a strictly fair comparison so that we can study whether optimizing the parameters in encoder $\phi_{\theta}(x)$ makes a difference. The perceived conflict stems from our pedagogical choice in Section 2.2, which we will clarify in the final manuscript:
> > >
> > > - Linear kernel as exposition: In Section 2.2, we introduced the linear kernel primarily to establish a clear mathematical bridge between Case I (empirical statistics) and Case II (learned embeddings). By using a linear formulation, we could demonstrate how the sample covariance is equivalent to linear kernel with scores as features.
> > >
> > > - Flexibility of the framework: As noted in our Remark in Section 2.2, Case II of the ProEval framework is explicitly designed to be kernel-agnostic and it also applies to the Matérn kernel $k(x, x') = k_{\text{Mat\'ern}}(\phi_\theta(x), \phi_{\theta}(x'))$. The core of our transfer learning "Case II" is the optimization of the log likelihood over parameters $\theta$ of the encoder $\phi_{\theta}(x)$.
> > >
> > > - Interpretation of results:  Table 1 and Table 2 represent fundamentally different transfer scenarios. Table 1 (Case I) relies on historical data from the same questions, while Table 2 (Case II) evaluates zero-shot generalization to entirely new datasets using the encoder structure shown in Figure 3. The use of the Matérn kernel in Table 2 does not change our primary conclusion: optimizing the encoder via Case II transfer boosts BQ performance.
> > >
> > >
> > > > W5: Query generator validity
> > >
> > > We acknowledge your concern that the reported failure rates could be influenced by mistakes made by the query generator when synthesizing "harder" problems. We emphasize that this is exactly the reason that we chose the strongest model at the time, Gemini 3 Pro, as the generator, and it is aligned with the widely used LLM-as-a-judge setup.
> > >
> > > - Relative comparison validity: All methods in our synthesis experiments (Rand-Gen, SS-Gen, TSS) utilize the same query generator under identical temperature settings. Therefore, the relative performance delta remains a valid indicator of ProEval’s ability to surface model-specific vulnerabilities.
> > >
> > > - Efficiency vs. ground truth: The focus of ProEval is the efficient discovery of potential failure regions. Even if some discovered cases are "false positives" due to generator error, ProEval significantly reduces the search space for human auditors compared to random exploration.
> > >
> > > - Commitment to human audit: We agree that a human audit strengthens the paper’s significance. To quantify the ground-truth accuracy of the query generator, we conducted a human verification study on 80 randomly sampled questions from the failure discovery task (generated by Rand-Gen, SS-Gen and TSS), covering grade school math (GSM8K-style) and implicit reasoning (StrategyQA-style).
> > >
> > > We found that Gemini 3 Pro answered 90% of these questions correctly. For the 8 questions it answered incorrectly, we inspected the answers given by the target model Gemini 2.5 Flash. On 5 of those questions, the target model gave the same incorrect answer as Gemini 3 Pro; on 2 questions, it gave a different but still incorrect answer; and on 1 question (where the answer should be yes or no), it gave the correct answer but used flawed reasoning. Overall, this human study demonstrates that our estimated failure rate is a lower bound of the true failure rate.
> > >
> > > Intuitively, this makes sense because a “weaker” model will very likely fail on problems that a “stronger” model failed at. We will add this human study and detailed analyses to the paper.
> > >
> > > We believe these clarifications and the addition of a human audit directly address your remaining concerns. We will ensure these details are explicitly stated in Sections 2.2 and 3.3 of the revised paper.

---

### Official Review · Reviewer_h2vp · 2026-03-12

**Soundness:** 4
**Presentation:** 3
**Significance:** 4
**Originality:** 3
**Overall Recommendation:** 5
**Confidence:** 4

**Summary:**

In this paper, authors present ProEval, a proactive framework for evaluating Generative AI (GenAI) models more efficiently. Evaluating these models is traditionally resource-intensive due to the high costs for human or LLM-based raters and massive scale of modern benchmarks. ProEval addresses these challenges by using transfer learning and Gaussian Processes (GPs) to estimate model performance and discover failures with significantly fewer samples than standard methods.

**Compliance With Llm Reviewing Policy:**

Affirmed.

**Key Questions For Authors:**

1. How reliable is knowledge transfer across benchmarks? Particularly, if the tasks are very different.  How effectively can the GP transfer knowledge between different modalities (e.g., using text evaluation results to inform image generation performance)? It would be good to show a few qualitative examples.

2. How would the framework handle models that are specifically fine-tuned to "break" common semantic correlations identified in historical benchmarks?

3. Your evaluations cover general toxicity; how does ProEval perform on highly subjective or context-dependent safety categories, such as subtle political bias or specific legal compliance?

**Limitations:**

yes

**Strengths And Weaknesses:**

Strengths
1. The paper is technically robust and provides theoretical derivations with extensive empirical validation.

2. ProEval introduces several novel technical approaches to evaluation. Particularly, active topic-aware data synthesis represents a shift from "testing" to "red-teaming" for performance estimation.

3. The paper provides a timely, theoretically grounded, and empirically superior solution to the growing costs of GenAI evaluation.

Weaknesses

1.  The efficacy of the framework relies heavily on the quality and relevance of the GP prior. If a target model has low correlation with historical source models, the performance estimation accuracy drops significantly. This is evidenced by models like G0 and O1, which were flagged as outliers because they lacked sufficient correlation with the source data.

2. For cross-benchmark evaluation (Case II), success depends on the ability of the chosen embedding model to capture semantic similarities related to input difficulty. The authors acknowledge that future work is needed to reduce reliance on these high-quality embeddings

---

> ### Author Rebuttal · Authors · 2026-03-30
>
> Thank you for your insightful comments.
>
>
> > W1: The efficacy of the framework relies heavily on the quality and relevance of the GP prior..
>
>
>
>
>
>
> We contend that leveraging strong GP priors is a core advantage of our approach rather than a weakness. In the absence of such priors, a surrogate model would be forced to learn from scratch through direct observations of the function $f$, which is undesirable given that querying $f$ is highly expensive. Additionally, as noted regarding G0 and O1, ProEval incorporates a mechanism to evaluate the quality of available priors; by verifying the sufficiency of source data, the system can strategically abstain from making predictions when a reliable prior is missing. We will ensure these advantages are articulated more clearly in the revised manuscript.
>
>
> > W2: reliance on high-quality embedding
>
>
> In our continued investigation, we tested four embedding models (Google gemini_embedding_001, OpenAI text_embedding_3_small, OpenAI text_embedding_3_large, and all_minilm_l6_v2), using a sampling budget of 50. We evaluated Gemini 2.5 Flash on StrategyQA using BQ Vanilla, the result shows a clear trend: models with larger embedding dimensions tend to yield lower MAEs.
>
>
> | Embedding Model | Dimension | Tier | BQ MAE |
> | :--- | ---: | :--- | -: |
> | `gemini_embedding_001` | 3072 | Strong | 0.0425 |
> | `text_embedding_3_large` | 3072 | Strong | 0.0654 |
> | `text_embedding_3_small` | 1536 | Medium | 0.0834 |
> | `all_minilm_l6_v2` | 384 | Weak | 0.0900 |
>
>
> We believe these new insights will be valuable foundations for future research.
>
>
> > Q1: How reliable is knowledge transfer across benchmarks? Particularly, if the tasks are very different. How effectively can the GP transfer knowledge between different modalities (e.g., using text evaluation results to inform image generation performance)? It would be good to show a few qualitative examples.
>
>
> We performed the following experiments for BQ to investigate transferring across modalities. We used DICES (text) dataset to perform the knowledge transfer, and ran ProEval on DIVE (image) data with 15 samples. The results show that, compared with the no knowledge transfer, the cross-modality knowledge contributes significantly, reducing the MAE from 0.111 to 0.055.
>
>
> > Q2: How would the framework handle models that are specifically fine-tuned to "break" common semantic correlations identified in historical benchmarks?
>
>
> We are not entirely sure whether the reviewer is referring to embedding models (in the GP) or target models (the ones we are evaluating). If it is the embedding model, ProEval handles it by learning an encoder that takes the embedding model’s output as the input, so that it can ensure the GP will still fit the historical data. If it is the target model, ProEval handles this issue by adopting the GP based on empirical statistics (case I), and it does not depend on the semantic correlations.
>
>
> > Q3: ..how does ProEval perform on highly subjective or context-dependent safety categories, such as subtle political bias or specific legal compliance?
>
>
> If we understand correctly, the question is about how ProEval handles the subjective judgements in safety. The DICES dataset includes the ratings from diverse raters that have different opinions. We aggregate the ratings from all raters on one example (x) and set it as the GP observation f(x). The results can be found in Table 1.

---

> > ### Author Rebuttal · Reviewer_h2vp · 2026-04-06
> >
> > I will keep the positive score.

---

### Decision · Program_Chairs · 2026-04-30

**Decision:**

Accept (regular)

**Comment:**

Reviewers agreed that this submission addresses a timely and practically important problem, reducing the cost of GenAI evaluation, and praised the principled GP/Bayesian quadrature formulation, the unified treatment of performance estimation and failure discovery, and the breadth of experiments across models, benchmarks, and transfer settings. The main concerns centered on the method’s reliance on informative priors and strong embeddings, the behavior when transfer assumptions fail, theoretical and presentation inconsistencies around normalization and bounds, ambiguity about kernel choices in Case II, and the possibility that failure-discovery rates are influenced by query-generator errors. The author rebuttal was substantive and improved consensus: it clarified the normalization typo and revised proof framing, explained source-model filtering and the intended scope of the assumptions, added evidence on embedding sensitivity and cross-modality transfer, and reported a human audit supporting the validity of synthetic failure cases. Some limitations remain: notably residual dependence on prior quality and the need for clearer final exposition of kernel choices and generator-related caveats. However,  these do not seem to undermine the main claims.